# Gradient Surgery for Multi-Task Learning

**Tianhe Yu**[1], **Saurabh Kumar**[1], **Abhishek Gupta**[2], **Sergey Levine**[2],
**Karol Hausman**[3], **Chelsea Finn**[1]
Stanford University[1], UC Berkeley[2], Robotics at Google[3]
`tianheyu@cs.stanford.edu`

## Abstract

While deep learning and deep reinforcement learning (RL) systems have demonstrated impressive results in domains such as image classification, game playing, and robotic control, data efficiency remains a major challenge. Multi-task learning has emerged as a promising approach for sharing structure across multiple tasks to enable more efficient learning. However, the multi-task setting presents a number of optimization challenges, making it difficult to realize large efficiency gains compared to learning tasks independently. The reasons why multi-task learning is so challenging compared to single-task learning are not fully understood. In this work, we identify a set of three conditions of the multi-task optimization landscape that cause detrimental gradient interference, and develop a simple yet general approach for avoiding such interference between task gradients. We propose a form of gradient surgery that projects a task's gradient onto the normal plane of the gradient of any other task that has a *conflicting* gradient. On a series of challenging multi-task supervised and multi-task RL problems, this approach leads to substantial gains in efficiency and performance. Further, it is model-agnostic and can be combined with previously-proposed multi-task architectures for enhanced performance.

## 1   Introduction

While deep learning and deep reinforcement learning (RL) have shown considerable promise in enabling systems to learn complex tasks, the data requirements of current methods make it difficult to learn a breadth of capabilities, particularly when all tasks are learned individually from scratch. A natural approach to such multi-task learning problems is to train a network on all tasks jointly, with the aim of discovering shared structure across the tasks in a way that achieves greater efficiency and performance than solving tasks individually. However, learning multiple tasks all at once results is a difficult optimization problem, sometimes leading to *worse* overall performance and data efficiency compared to learning tasks individually [42, 50]. These optimization challenges are so prevalent that multiple multi-task RL algorithms have considered using independent training as a subroutine of the algorithm before distilling the independent models into a multi-tasking model [32, 42, 50, 21, 56], producing a multi-task model but losing out on the efficiency gains over independent training. If we could tackle the optimization challenges of multi-task learning effectively, we may be able to actually realize the hypothesized benefits of multi-task learning without the cost in final performance.

While there has been a significant amount of research in multi-task learning [6, 49], the optimization challenges are not well understood. Prior work has described varying learning speeds of different tasks [8, 26] and plateaus in the optimization landscape [52] as potential causes, whereas a range of other works have focused on the model architecture [40, 33]. In this work, we instead hypothesize that one of the main optimization issues in multi-task learning arises from gradients from different tasks conflicting with one another in a way that is detrimental to making progress. We define two gradients to be conflicting if they point away from one another, i.e., have a negative cosine similarity. We hypothesize that such conflict is detrimental when a) conflicting gradients coincide with b) high positive curvature and c) a large difference in gradient magnitudes.

As an illustrative example, consider the 2D optimization landscapes of two task objectives in Figure 1a-c. The optimization landscape of each task consists of a deep valley, a property that has been observed in neural network optimization landscapes [22], and the bottom of each valley is characterized by high positive curvature and large differences in the task gradient magnitudes. Under such circumstances, the multi-task gradient is dominated by one task gradient, which comes at the cost of degrading the performance of the other task. Further, due to high

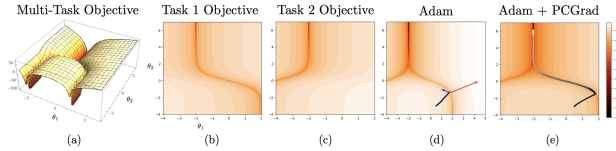

Figure 1: Visualization of PCGrad on a 2D multi-task optimization problem. (a) A multi-task objective landscape. (b) & (c) Contour plots of the individual task objectives that comprise (a). (d) Trajectory of gradient updates on the multi-task objective using the Adam optimizer. The gradient vectors of the two tasks at the end of the trajectory are indicated by blue and red arrows, where the relative lengths are on a log scale.(e) Trajectory of gradient updates on the multi-task objective using Adam with PCGrad. For (d) and (e), the optimization trajectory goes from black to yellow.

curvature, the improvement in the dominating task may be overestimated, while the degradation in performance of the non-dominating task may be underestimated. As a result, the optimizer struggles to make progress on the optimization objective. In Figure 1d), the optimizer reaches the deep valley of task 1, but is unable to traverse the valley in a parameter setting where there are *conflicting gradients*, *high curvature*, and *a large difference in gradient magnitudes* (see gradients plotted in Fig. 1d). In Section 5.3, we find experimentally that this *tragic triad* also occurs in a higher-dimensional neural network multi-task learning problem.

The core contribution of this work is a method for mitigating gradient interference by altering the gradients directly, i.e. by performing "gradient surgery." If two gradients are conflicting, we alter the gradients by projecting each onto the normal plane of the other, preventing the interfering components of the gradient from being applied to the network. We refer to this particular form of gradient surgery as *projecting conflicting gradients* (PCGrad). PCGrad is model-agnostic, requiring only a single modification to the application of gradients. Hence, it is easy to apply to a range of problem settings, including multi-task supervised learning and multi-task reinforcement learning, and can also be readily combined with other multi-task learning approaches, such as those that modify the architecture. We theoretically prove the local conditions under which PCGrad improves upon standard multi-task gradient descent, and we empirically evaluate PCGrad on a variety of challenging problems, including multi-task CIFAR classification, multi-objective scene understanding, a challenging multi-task RL domain, and goal-conditioned RL. Across the board, we find PCGrad leads to substantial improvements in terms of data efficiency, optimization speed, and final performance compared to prior approaches, including a more than 30% absolute improvement in multi-task reinforcement learning problems. Further, on multi-task supervised learning tasks, PCGrad can be successfully combined with prior state-of-the-art methods for multi-task learning for even greater performance.

## 2 Multi-Task Learning with PCGrad

While the multi-task problem can in principle be solved by simply applying a standard single-task algorithm with a suitable task identifier provided to the model, or a simple multi-head or multi-output model, a number of prior works [42, 50, 53] have found this learning problem to be difficult. In this section, we introduce notation, identify possible causes for the difficulty of multi-task optimization, propose a simple and general approach to mitigate it, and theoretically analyze the proposed approach.

### 2.1 Preliminaries: Problem and Notation

The goal of multi-task learning is to find parameters $\theta$ of a model $f_\theta$ that achieve high average performance across all the training tasks drawn from a distribution of tasks $p(\mathcal{T})$. More formally, we aim to solve the problem: $\min_\theta \mathbb{E}_{\mathcal{T}_i \sim p(\mathcal{T})}\left[\mathcal{L}_i(\theta)\right]$, where $\mathcal{L}_i$ is a loss function for the $i$-th task $\mathcal{T}_i$ that we want to minimize. For a set of tasks, $\{\mathcal{T}_i\}$, we denote the multi-task loss as $\mathcal{L}(\theta) = \sum_i \mathcal{L}_i(\theta)$, and the gradients of each task as $\mathbf{g}_i = \nabla \mathcal{L}_i(\theta)$ for a particular $\theta$. (We drop the reliance on $\theta$ in the notation for brevity.) To obtain a model that solves a specific task from the task distribution $p(\mathcal{T})$, we define a task-conditioned model $f_\theta(y|x, z_i)$, with input $x$, output $y$, and encoding $z_i$ for task $\mathcal{T}_i$, which could be provided as a one-hot vector or in any other form.

## 2.2 The Tragic Triad: Conflicting Gradients, Dominating Gradients, High Curvature

We hypothesize that a key optimization issue in multi-task learning arises from conflicting gradients, where gradients for different tasks point away from one another as measured by a negative inner product. However, conflicting gradients are not detrimental on their own. Indeed, simply averaging task gradients should provide the correct solution to descend the multi-task objective. However, there are conditions under which such conflicting gradients lead to significantly degraded performance. Consider a two-task optimization problem. If the gradient of one task is much larger in magnitude than the other, it will dominate the average gradient. If there is also high positive curvature along the directions of the task gradients, then the improvement in performance from the dominating task may be significantly overestimated, while the degradation in performance from the dominated task may be significantly underestimated. Hence, we can characterize the co-occurrence of three conditions as follows: (a) when gradients from multiple tasks are in conflict with one another (b) when the difference in gradient magnitudes is large, leading to some task gradients dominating others, and (c) when there is high curvature in the multi-task optimization landscape. We formally define the three conditions below.

**Definition 1.** *We define $\phi_{ij}$ as the angle between two task gradients $\mathbf{g}_i$ and $\mathbf{g}_j$. We define the gradients as **conflicting** when $\cos\phi_{ij} < 0$.*

**Definition 2.** *We define the **gradient magnitude similarity** between two gradients $\mathbf{g}_i$ and $\mathbf{g}_j$ as $\Phi(\mathbf{g}_i, \mathbf{g}_j) = \frac{2\|\mathbf{g}_i\|_2\|\mathbf{g}_j\|_2}{\|\mathbf{g}_i\|_2^2 + \|\mathbf{g}_j\|_2^2}$.*

When the magnitude of two gradients is the same, this value is equal to 1. As the gradient magnitudes become increasingly different, this value goes to zero.

**Definition 3.** *We define **multi-task curvature** as $\mathbf{H}(\mathcal{L}; \theta, \theta') = \int_0^1 \nabla\mathcal{L}(\theta)^T \nabla^2\mathcal{L}(\theta + a(\theta' - \theta))\nabla\mathcal{L}(\theta)da$, which is the averaged curvature of $\mathcal{L}$ between $\theta$ and $\theta'$ in the direction of the multi-task gradient $\nabla\mathcal{L}(\theta)$.*

When $\mathbf{H}(\mathcal{L}; \theta, \theta') > C$ for some large positive constant $C$, for model parameters $\theta$ and $\theta'$ at the current and next iteration, we characterize the optimization landscape as having high curvature.

We aim to study the tragic triad and observe the presence of the three conditions through two examples. First, consider the two-dimensional optimization landscape illustrated in Fig. 1a, where the landscape for each task objective corresponds to a deep and curved valley with large curvatures (Fig. 1b and 1c). The optima of this multi-task objective correspond to where the two valleys meet. More details on the optimization landscape are in Appendix D. Particular points of this optimization landscape exhibit the three described conditions, and we observe that, the Adam [30] optimizer stalls precisely at one of these points (see Fig. 1d), preventing it from reaching an optimum. This provides some empirical evidence for our hypothesis. Our experiments in Section 5.3 further suggest that this phenomenon occurs in multi-task learning with deep networks. Motivated by these observations, we develop an algorithm that aims to alleviate the optimization challenges caused by conflicting gradients, dominating gradients, and high curvature, which we describe next.

## 2.3 PCGrad: Project Conflicting Gradients

Our goal is to break one condition of the tragic triad by directly altering the gradients themselves to prevent conflict. In this section, we outline our approach for altering the gradients. In the next section, we will theoretically show that de-conflicting gradients can benefit multi-task learning when dominating gradients and high curvatures are present.

To be maximally effective and widely applicable, we aim to alter the gradients in a way that allows for positive interactions between the task gradients and does not introduce assumptions on the form of the model. Hence, when gradients do not conflict, we do not change the gradients. When gradients do conflict, the goal of PCGrad is to modify the gradients for each task so as to minimize negative conflict with other task gradients, which will in turn mitigate under- and over-estimation problems arising from high curvature.

To deconflict gradients during optimization, PCGrad adopts a simple procedure: if the gradients between two tasks are in conflict, i.e. their cosine similarity is negative, we project the gradient of each task onto the normal plane of the gradient of the other task. This amounts to removing the conflicting component of the gradient for the task, thereby reducing the amount of destructive gradient interference between tasks. A pictorial description of this idea is shown in Fig. 2.

**Algorithm 1** PCGrad Update Rule

**Require:** Model parameters $\theta$, task minibatch $\mathcal{B} = \{\mathcal{T}_k\}$

1: $\mathbf{g}_k \leftarrow \nabla_\theta \mathcal{L}_k(\theta) \ \forall k$
2: $\mathbf{g}_k^{\text{PC}} \leftarrow \mathbf{g}_k \ \forall k$
3: **for** $\mathcal{T}_i \in \mathcal{B}$ **do**
4:     **for** $\mathcal{T}_j \overset{\text{uniformly}}{\sim} \mathcal{B} \setminus \mathcal{T}_i$ in random order **do**
5:         **if** $\mathbf{g}_i^{\text{PC}} \cdot \mathbf{g}_j < 0$ **then**
6:             // Subtract the projection of $\mathbf{g}_i^{\text{PC}}$ onto $\mathbf{g}_j$
7:             Set $\mathbf{g}_i^{\text{PC}} = \mathbf{g}_i^{\text{PC}} - \frac{\mathbf{g}_i^{\text{PC}} \cdot \mathbf{g}_j}{\|\mathbf{g}_j\|^2} \mathbf{g}_j$
8: **return** update $\Delta\theta = \mathbf{g}^{\text{PC}} = \sum_i \mathbf{g}_i^{\text{PC}}$

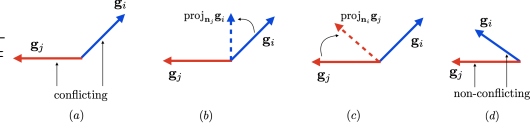

Figure 2: Conflicting gradients and PCGrad. In (a), tasks $i$ and $j$ have conflicting gradient directions, which can lead to destructive interference. In (b) and (c), we illustrate the PCGrad algorithm in the case where gradients are conflicting. PCGrad projects task $i$'s gradient onto the normal vector of task $j$'s gradient, and vice versa. Non-conflicting task gradients (d) are not altered under PCGrad, allowing for constructive interaction.

Suppose the gradient for task $\mathcal{T}_i$ is $\mathbf{g}_i$, and the gradient for task $\mathcal{T}_j$ is $\mathbf{g}_j$. PCGrad proceeds as follows: (1) First, it determines whether $\mathbf{g}_i$ conflicts with $\mathbf{g}_j$ by computing the cosine similarity between vectors $\mathbf{g}_i$ and $\mathbf{g}_j$, where negative values indicate conflicting gradients. (2) If the cosine similarity is negative, we replace $\mathbf{g}_i$ by its projection onto the normal plane of $\mathbf{g}_j$: $\mathbf{g}_i = \mathbf{g}_i - \frac{\mathbf{g}_i \cdot \mathbf{g}_j}{\|\mathbf{g}_j\|^2} \mathbf{g}_j$. If the gradients are not in conflict, i.e. cosine similarity is non-negative, the original gradient $\mathbf{g}_i$ remains unaltered. (3) PCGrad repeats this process across all of the other tasks sampled in random order from the current batch $\mathcal{T}_j \ \forall \ j \neq i$, resulting in the gradient $\mathbf{g}_i^{\text{PC}}$ that is applied for task $\mathcal{T}_i$. We perform the same procedure for all tasks in the batch to obtain their respective gradients. The full update procedure is described in Algorithm 1 and a discussion on using a random task order is included in Appendix H.

This procedure, while simple to implement, ensures that the gradients that we apply for each task per batch interfere minimally with the other tasks in the batch, mitigating the conflicting gradient problem, producing a variant on standard first-order gradient descent in the multi-objective setting. In practice, PCGrad can be combined with any gradient-based optimizer, including commonly used methods such as SGD with momentum and Adam [30], by simply passing the computed update to the respective optimizer instead of the original gradient. Our experimental results verify the hypothesis that this procedure reduces the problem of conflicting gradients, and find that, as a result, learning progress is substantially improved.

## 2.4 Theoretical Analysis of PCGrad

In this section, we theoretically analyze the performance of PCGrad with two tasks:

**Definition 4.** *Consider two task loss functions $\mathcal{L}_1 : \mathbb{R}^n \to \mathbb{R}$ and $\mathcal{L}_2 : \mathbb{R}^n \to \mathbb{R}$. We define the two-task learning objective as $\mathcal{L}(\theta) = \mathcal{L}_1(\theta) + \mathcal{L}_2(\theta)$ for all $\theta \in \mathbb{R}^n$, where $\mathbf{g}_1 = \nabla\mathcal{L}_1(\theta)$, $\mathbf{g}_2 = \nabla\mathcal{L}_2(\theta)$, and $\mathbf{g} = \mathbf{g}_1 + \mathbf{g}_2$.*

We first aim to verify that the PCGrad update corresponds to a sensible optimization procedure under simplifying assumptions. We analyze convergence of PCGrad in the convex setting, under standard assumptions in Theorem 1. For additional analysis on convergence, including the non-convex setting, with more than two tasks, and with momentum-based optimizers, see Appendices A.1 and A.4

**Theorem 1.** *Assume $\mathcal{L}_1$ and $\mathcal{L}_2$ are convex and differentiable. Suppose the gradient of $\mathcal{L}$ is $L$-Lipschitz with $L > 0$. Then, the PCGrad update rule with step size $t \leq \frac{1}{L}$ will converge to either (1) a location in the optimization landscape where $\cos(\phi_{12}) = -1$ or (2) the optimal value $\mathcal{L}(\theta^*)$.*

*Proof.* See Appendix A.1. □

Theorem 1 states that application of the PCGrad update in the two-task setting with a convex and Lipschitz multi-task loss function $\mathcal{L}$ leads to convergence to either the minimizer of $\mathcal{L}$ or a potentially sub-optimal objective value. A sub-optimal solution occurs when the cosine similarity between the gradients of the two tasks is exactly $-1$, i.e. the gradients directly conflict, leading to zero gradient after applying PCGrad. However, in practice, since we are using SGD, which is a noisy estimate of the true batch gradients, the cosine similarity between the gradients of two tasks in a minibatch is unlikely to be $-1$, thus avoiding this scenario. Note that, in theory, convergence may be slow if $\cos(\phi_{12})$ hovers near $-1$. However, we don't observe this in practice, as seen in the objective-wise learning curves in Appendix B.

Now that we have checked the sensibility of PCGrad, we aim to understand how PCGrad relates to the three conditions in the tragic triad. In particular, we derive sufficient conditions under which PCGrad achieves lower loss after one update. Here, we still analyze the two task setting, but no longer assume convexity of the loss functions.

**Definition 5.** *We define the **multi-task curvature bounding measure** $\xi(\mathbf{g}_1, \mathbf{g}_2) = (1 - \cos^2 \phi_{12}) \frac{\|\mathbf{g}_1 - \mathbf{g}_2\|_2^2}{\|\mathbf{g}_1 + \mathbf{g}_2\|_2^2}.$*

With the above definition, we present our next theorem:

**Theorem 2.** *Suppose $\mathcal{L}$ is differentiable and the gradient of $\mathcal{L}$ is Lipschitz continuous with constant $L > 0$. Let $\theta^{MT}$ and $\theta^{PCGrad}$ be the parameters after applying one update to $\theta$ with $\mathbf{g}$ and PCGrad-modified gradient $\mathbf{g}^{PC}$ respectively, with step size $t > 0$. Moreover, assume $\mathbf{H}(\mathcal{L}; \theta, \theta^{MT}) \geq \ell \|\mathbf{g}\|_2^2$ for some constant $\ell \leq L$, i.e. the multi-task curvature is lower-bounded. Then $\mathcal{L}(\theta^{PCGrad}) \leq \mathcal{L}(\theta^{MT})$ if (a) $\cos \phi_{12} \leq -\Phi(\mathbf{g}_1, \mathbf{g}_2)$, (b) $\ell \geq \xi(\mathbf{g}_1, \mathbf{g}_2)L$, and (c) $t \geq \frac{2}{\ell - \xi(\mathbf{g}_1, \mathbf{g}_2)L}$.*

*Proof.* See Appendix A.2. □

Intuitively, Theorem 2 implies that PCGrad achieves lower loss value after a single gradient update compared to standard gradient descent in multi-task learning when (i) the angle between task gradients is not too small, i.e. the two tasks need to conflict sufficiently (**condition (a)**), (ii) the difference in magnitude needs to be sufficiently large (**condition (a)**), (iii) the curvature of the multi-task gradient should be large (**condition (b)**), (iv) and the learning rate should be big enough so that large curvature would lead to overestimation of performance improvement on the dominating task and underestimation of performance degradation on the dominated task (**condition (c)**). These first three points (i-iii) correspond to exactly the triad of conditions outlined in Section 2.2, while the latter condition (iv) is desirable as we hope to learn quickly. We empirically validate that the first three points, (i-iii), are frequently met in a neural network multi-task learning problem in Figure 4 in Section 5.3. For additional analysis, including complete sufficient and necessary conditions for the PCGrad update to outperform the vanilla multi-task gradient, see Appendix A.3.

# 3 PCGrad in Practice

We use PCGrad in supervised learning and reinforcement learning problems with multiple tasks or goals. Here, we discuss the practical application of PCGrad to those settings.

In multi-task supervised learning, each task $\mathcal{T}_i \sim p(\mathcal{T})$ has a corresponding training dataset $\mathcal{D}_i$ consisting of labeled training examples, i.e. $\mathcal{D}_i = \{(x, y)_n\}$. The objective for each task in this supervised setting is then defined as $\mathcal{L}_i(\theta) = \mathbb{E}_{(x,y) \sim \mathcal{D}_i} [-\log f_\theta(y|x, z_i)]$, where $z_i$ is a one-hot encoding of task $\mathcal{T}_i$. At each training step, we randomly sample a batch of data points $\mathcal{B}$ from the whole dataset $\bigcup_i \mathcal{D}_i$ and then group the sampled data with the same task encoding into small batches denoted as $\mathcal{B}_i$ for each $\mathcal{T}_i$ represented in $\mathcal{B}$. We denote the set of tasks appearing in $\mathcal{B}$ as $\mathcal{B}_\mathcal{T}$. After sampling, we precompute the gradient of each task in $\mathcal{B}_\mathcal{T}$ as $\nabla_\theta \mathcal{L}_i(\theta) = \mathbb{E}_{(x,y) \sim \mathcal{B}_i} [-\nabla_\theta \log f_\theta(y|x, z_i)]$. Given the set of precomputed gradients $\nabla_\theta \mathcal{L}_i(\theta)$, we also precompute the cosine similarity between all pairs of the gradients in the set. Using the pre-computed gradients and their similarities, we can obtain the PCGrad update by following Algorithm 1, without re-computing task gradients nor backpropagating into the network. Since the PCGrad procedure is only modifying the gradients of shared parameters in the optimization step, it is *model-agnostic* and can be applied to any architecture with shared parameters. We empirically validate PCGrad with multiple architectures in Section 5.

For multi-task RL and goal-conditioned RL, PCGrad can be readily applied to policy gradient methods by directly updating the computed policy gradient of each task, following Algorithm 1, analogous to the supervised learning setting. For actor-critic algorithms, it is also straightforward to apply PCGrad: we simply replace task gradients for both the actor and the critic by their gradients computed via PCGrad. For more details on the practical implementation for RL, see Appendix C.

# 4 Related Work

Algorithms for multi-task learning typically consider how to train a single model that can solve a variety of different tasks [6, 2, 49]. The multi-task formulation has been applied to many different settings, including supervised learning [63, 35, 60, 53, 62] and reinforcement-learning [17, 58], as well as many different domains, such as vision [3, 39, 31, 33, 62], language [11, 15, 38, 44] and robotics [45, 59, 25]. While multi-task learning has the promise of accelerating acquisition of large

task repertoires, in practice it presents a challenging optimization problem, which has been tackled in several ways in prior work.

A number of architectural solutions have been proposed to the multi-task learning problem based on multiple modules or paths [19, 14, 40, 51, 46, 57, 46], or using attention-based architectures [33, 37]. Our work is agnostic to the model architecture and can be combined with prior architectural approaches in a complementary fashion. A different set of multi-task learning approaches aim to decompose the problem into multiple local problems, often corresponding to each task, that are significantly easier to learn, akin to divide and conquer algorithms [32, 50, 42, 56, 21, 13]. Eventually, the local models are combined into a single, multi-task policy using different distillation techniques (outlined in [27, 13]). In contrast to these methods, we propose a simple and cogent scheme for multi-task learning that allows us to learn the tasks simultaneously using a single, shared model without the need for network distillation.

Similarly to our work, a number of prior approaches have observed the difficulty of optimization in multi-task learning [26, 29, 52, 55]. Our work suggests that the challenge in multi-task learning may be attributed to what we describe as the tragic triad of multi-task learning (i.e., conflicting gradients, high curvature, and large gradient differences), which we address directly by introducing a simple and practical algorithm that deconflicts gradients from different tasks. Prior works combat optimization challenges by rescaling task gradients [53, 9]. We alter both the magnitude and direction of the gradient, which we find to be critical for good performance (see Fig. 3). Prior work has also used the cosine similarity between gradients to define when an auxiliary task might be useful [16] or when two tasks are related [55]. We similarly use cosine similarity between gradients to determine if the gradients between a pair of tasks are in conflict. Unlike Du et al. [16], we use this measure for effective multi-task learning, instead of ignoring auxiliary objectives. Overall, we empirically compare our approach to a number of these prior approaches [53, 9, 55], and observe superior performance with PCGrad.

Multiple approaches to continual learning have studied how to prevent gradient updates from adversely affecting previously-learned tasks through various forms of gradient projection [36, 7, 18, 23]. These methods focus on sequential learning settings, and solve for the gradient projections using quadratic programming [36], only project onto the normal plane of the average gradient of past tasks [7], or project the current task gradients onto the orthonormal set of previous task gradients [18]. In contrast, our work focuses on positive transfer when simultaneously learning multiple tasks, does not require solving a QP, and *iteratively* projects the gradients of each task instead of *averaging* or only projecting the *current* task gradient. Finally, our method is distinct from and solves a different problem than the projected gradient method [5], which is an approach for constrained optimization that projects gradients onto the constraint manifold.

# 5   Experiments

The goal of our experiments is to study the following questions: (1) Does PCGrad make the optimization problems easier for various multi-task learning problems including supervised, reinforcement, and goal-conditioned reinforcement learning settings across different task families? (2) Can PCGrad be combined with other multi-task learning approaches to further improve performance? (3) Is the tragic triad of multi-task learning a major factor in making optimization for multi-task learning challenging? To broadly evaluate PCGrad, we consider multi-task supervised learning, multi-task RL, and goal-conditioned RL problems. We include the results on goal-conditioned RL in Appendix F.

During our evaluation, we tune the parameters of the baselines independently, ensuring that all methods were fairly provided with equal model and training capacity. PCGrad inherits the hyperparameters of the respective baseline method in all experiments, and has no additional hyperparameters. For more details on the experimental set-up and model architectures, see Appendix J. The code is available online[1].

## 5.1   Multi-Task Supervised Learning

To answer question (1) in the supervised learning setting and question (2), we perform experiments on five standard multi-task supervised learning datasets: MultiMNIST, CityScapes, CelebA, multi-task CIFAR-100 and NYUv2. We include the results on MultiMNIST and CityScapes in Appendix E.

| #P. | Architecture | Weighting | Segmentation (Higher Better) | | Depth (Lower Better) | | Surface Normal Angle Distance (Lower Better) | | Within $t°$ (Higher Better) | | |
|---|---|---|---|---|---|---|---|---|---|---|---|
| | | | mIoU | Pix Acc | Abs Err | Rel Err | Mean | Median | 11.25 | 22.5 | 30 |
| $\approx 3$ | Cross-Stitch[‡] | Equal Weights | 14.71 | 50.23 | 0.6481 | 0.2871 | 33.56 | 28.58 | 20.08 | 40.54 | 51.97 |
| | | Uncert. Weights* | 15.69 | 52.60 | 0.6277 | 0.2702 | 32.69 | 27.26 | 21.63 | 42.84 | 54.45 |
| | | DWA[†], $T=2$ | **16.11** | **53.19** | **0.5922** | **0.2611** | **32.34** | **26.91** | **21.81** | **43.14** | **54.92** |
| 1.77 | MTAN[†] | Equal Weights | **17.72** | 55.32 | **0.5906** | 0.2577 | 31.44 | **25.37** | 23.17 | 45.65 | 57.48 |
| | | Uncert. Weights* | 17.67 | **55.61** | 0.5927 | 0.2592 | **31.25** | 25.57 | 22.99 | **45.83** | **57.67** |
| | | DWA[†], $T=2$ | 17.15 | 54.97 | 0.5956 | **0.2569** | 31.60 | 25.46 | 22.48 | 44.86 | 57.24 |
| 1.77 | MTAN[†]+ PCGrad (ours) | Uncert. Weights* | 20.17 | 56.65 | 0.5904 | 0.2467 | 30.01 | 24.83 | 22.28 | 46.12 | 58.77 |

Table 1: Three-task learning on the NYUv2 dataset: 13-class semantic segmentation, depth estimation, and surface normal prediction results. #P shows the total number of network parameters. We highlight the best performing combination of multi-task architecture and weighting in bold. The top validation scores for each task are annotated with boxes. The symbols indicate prior methods: *: [28], [†]: [33], [‡]: [40]. Performance of other methods as reported in Liu et al. [33].

For CIFAR-100, we follow Rosenbaum et al. [46] to treat 20 coarse labels in the dataset as distinct tasks, creating a dataset with 20 tasks, with 2500 training instances and 500 test instances per task. We combine PCGrad with a powerful multi-task learning architecture, routing networks [46, 47], by applying PCGrad only to the shared parameters. For the details of this comparison, see Appendix J.1. As shown in Table 2, applying PCGrad to a single network achieves 71% classification accuracy, which outperforms most of the prior methods such as cross-stitch [40] and independent training, sug-

| | % accuracy |
|---|---|
| task specific, 1-fc [46] | 42 |
| task specific, all-fc [46] | 49 |
| cross stitch, all-fc [40] | 53 |
| routing, all-fc + WPL [47] | 74.7 |
| independent | 67.7 |
| PCGrad (ours) | 71 |
| routing-all-fc + WPL + PCGrad (ours) | **77.5** |

Table 2: CIFAR-100 multi-task results. When combined with routing networks, PCGrad leads to a large improvement.

gesting that sharing representations across tasks is conducive for good performance. While routing networks achieve better performance than PCGrad on its own, they are complementary: combining PCGrad with routing networks leads to a 2.8% absolute improvement in test accuracy.

We also aim to use PCGrad to tackle a multi-label classfication problem, which is a commonly used benchmark for multi-task learning. In multi-label classification, given a set of attributes, the model needs to decide whether each attribute describes the input. Hence, it is essentially a binary classification problem for each attribute. We choose the CelebA dataset [34], which consists of 200K face images with 40 attributes. Since for each attribution, it is a binary classfication problem and thus we convert it to a 40-way multi-task learning problem following [53]. We use the same architecture as in [53].

We use the binary classification error averaged across all 40 tasks to evaluate the performance as in [53]. Similar to the MultiMNIST results, we compare PCGrad to Sener and Koltun [53] by rerunning the open-sourced code provided in [53]. As shown in Table 3, PCGrad outperforms Sener and Koltun [53], suggesting that PCGrad is effective in multi-label classification and can also improve multi-task supervised learning performance when the number of tasks is high.

| | average classification error |
|---|---|
| Sener and Koltun [53] | 8.95 |
| PCGrad (ours) | **8.69** |

Table 3: CelebA results. We show the average classification error across all 40 tasks in CelebA. PCGrad outperforms the prior method Sener and Koltun [53] in this dataset.

Finally, we combine PCGrad with another state-of-art multi-task learning algorithm, MTAN [33], and evaluate the performance on a more challenging indoor scene dataset, NYUv2, which contains 3 tasks: 13-class semantic segmentation, depth estimation, and surface normal prediction. We compare MTAN with PCGrad to a list of methods mentioned in Appendix J.1, where each method is trained with three different weighting schemes as in [33], equal weighting, weight uncertainty [28], and DWA [33]. We only run MTAN with PCGrad with weight uncertainty as we find weight uncertainty

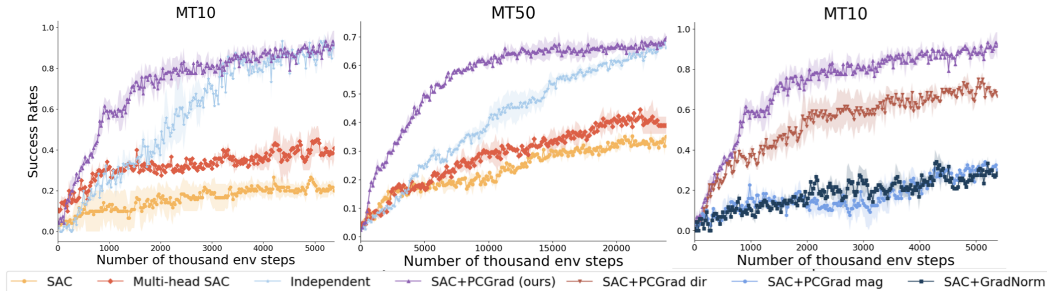

Figure 3: For the two plots on the left, we show learning curves on MT10 and MT50 respectively. PCGrad significantly outperforms the other methods in terms of both success rates and data efficiency. In the rightmost plot, we present the ablation study on only using the magnitude and the direction of gradients modified by PCGrad and a comparison to GradNorm [8]. PCGrad outperforms both ablations and GradNorm, indicating the importance of modifying both the gradient directions and magnitudes in multi-task learning.

as the most effective scheme for training MTAN. The results comparing Cross-Stitch, MTAN and MTAN + PCGrad are presented in Table 1 while the full comparison can be found in Table 8 in the Appendix J.4. MTAN with PCGrad is able to achieve the best scores in 8 out of the 9 categories where there are 3 categories per task.

Our multi-task supervised learning results indicate that PCGrad can be seamlessly combined with state-of-art multi-task learning architectures and further improve their results on established supervised multi-task learning benchmarks. We include more results of PCGrad combined with more multi-task learning architectures in Appendix I.

## 5.2  Multi-Task Reinforcement Learning

To answer question (2) in the RL setting, we first consider the multi-task RL problem and evaluate our algorithm on the recently proposed Meta-World benchmark [61]. In particular, we test all methods on the MT10 and MT50 benchmarks in Meta-World, which contain 10 and 50 manipulation tasks respectively shown in Figure 10. in Appendix J.2.

The results are shown in left two plots in Figure 3. PCGrad combined with SAC learns all tasks with the best data efficiency and successfully solves all of the 10 tasks in MT10 and about 70% of the 50 tasks in MT50. Training a single SAC policy and a multi-head policy is unable to acquire half of the skills in both MT10 and MT50, suggesting that eliminating gradient interference across tasks can significantly boost performance of multi-task RL. Training independent SAC agents is able to eventually solve all tasks in MT10 and 70% of the tasks in MT50, but requires about 2 millions and 15 millions more samples than PCGrad with SAC in MT10 and MT50 respectively, implying that applying PCGrad can result in leveraging shared structure among tasks that expedites multi-task learning. As noted by Yu et al. [61], these tasks involve distinct behavior motions, which makes learning all tasks with a single policy challenging as demonstrated by poor baseline performance. The ability to learn these tasks together opens the door for a number of interesting extensions to meta-learning and generalization to novel task families.

Since the PCGrad update affects both the gradient direction and the gradient magnitude, we perform an ablation study that tests two variants of PCGrad: (1) only applying the gradient direction corrected with PCGrad while keeping the gradient magnitude unchanged and (2) only applying the gradient magnitude computed by PCGrad while keeping the gradient direction unchanged. We further run a direction comparison to GradNorm [8], which also scales only the magnitudes of the task gradients. As shown in the rightmost plot in Figure 3, both variants and GradNorm perform worse than PCGrad and the variant where we only vary the gradient magnitude is much worse than PCGrad. This emphasizes the importance of the orientation change, which is particularly notable as multiple prior works only alter gradient magnitudes [8, 53]. We also notice that the variant of PCGrad where only the gradient magnitudes change achieves comparable performance to GradNorm, which suggests that it is important to modify both the gradient directions and magnitudes to eliminate interference and achieve good multi-task learning results. Finally, to test the importance of keeping positive cosine similarities between tasks for positive transfer, we compare PCGrad to a recently proposed method in [55] that regularizes cosine similarities of different task gradients towards 0. PCGrad outperforms Suteu and Guo [55] by a large margin. We leave details of the comparison to Appendix G.

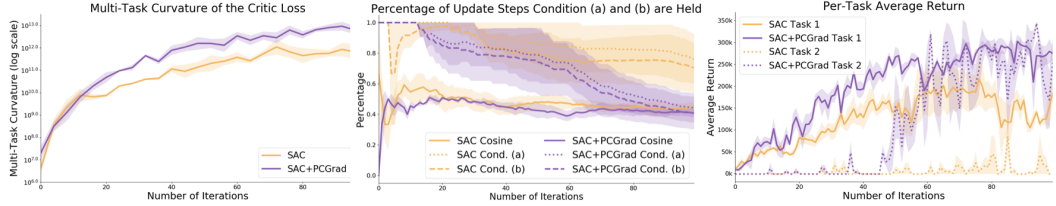

Figure 4: An empirical analysis of the theoretical conditions discussed in Theorem 2, showing the first 100 iterations of training on two RL tasks, reach and press button top. **Left**: The estimated value of the multi-task curvature. We observe high multi-task curvatures exist throughout training, providing evidence for condition (b) in Theorem 2. **Middle**: The solid lines show the percentage gradients with positive cosine similarity between two task gradients, while the dotted lines and dashed lines show the percentage of iterations in which condition (a) and the implication of condition (b) ($\xi(\mathbf{g}_1, \mathbf{g}_2) \leq 1$) in Theorem 2 are held respectively, among iterations when the cosine similarity is negative. **Right**: The average return of each task achieved by SAC and SAC combined with PCGrad. From the plots in the **Middle** and on the **Right**, we can tell that condition (a) holds most of the time for both Adam and Adam combined with PCGrad when they haven't solved Task 2 and as soon as Adam combined PCGrad starts to learn Task 2, the percentage of condition (a) held starts to decline. This observation suggests that condition (a) is a key factor for PCGrad excelling in multi-task learning.

## 5.3 Empirical Analysis of the Tragic Triad

Finally, to answer question (1), we compare the performance of standard multi-task SAC and the multi-task SAC with PCGrad. We evaluate each method on two tasks, reach and press button top, in the Meta-World [61] benchmark. As shown in the leftmost plot in Figure 4, we plot the multi-task curvature, which is computed as $\mathbf{H}(\mathcal{L}; \theta^t, \theta^{t+1}) = 2 \cdot \left[\mathcal{L}(\theta^{t+1}) - \mathcal{L}(\theta^t) - \nabla_{\theta^t}\mathcal{L}(\theta^t)^T(\theta^{t+1} - \theta^t)\right]$ by Taylor's Theorem where $\mathcal{L}$ is the multi-task loss, and $\theta^t$ and $\theta^{t+1}$ are the parameters at iteration $t$ and $t + 1$. During the training process, the multi-task curvature stays positive and is increasing for both Adam and Adam combined PCGrad, suggesting that condition (b) in Theorem 2 that the multi-task curvature is lower bounded by some positive value is widely held empirically. To further analyze conditions in Theorem 2 empirically, we plot the percentage of condition (a) (i.e. conflicting gradients) and the implication of condition (b) ($\xi(\mathbf{g}_1, \mathbf{g}_2) \leq 1$) in Theorem 2 being held among the total number of iterations where the cosine similarity is negative in the plot in the middle of Figure 4. Along with the plot on the right in Figure 4, which presents the average return of the two tasks during training, we can see that while Adam and Adam with PCGrad haven't received reward signal from Task 2, condition (a) and the implication of condition (b) stay held and as soon as Adam with PCGrad begins to solve Task 2, the percentage of condition (a) and the implication of condition (b) being held start to decrease. Such a pattern suggests that conflicting gradients, high curvatures and dominating gradients indeed produce considerable challenges in optimization before multi-task learner gains any useful learning signal, which also implies that the tragic triad may indeed be the determining factor where PCGrad can lead to better performance gain over standard multi-task learning in practice.

## 6 Conclusion

In this work, we identified a set of conditions that underlies major challenges in multi-task optimization: conflicting gradients, high positive curvature, and large gradient differences. We proposed a simple algorithm (PCGrad) to mitigate these challenges via "gradient surgery." PCGrad provides a simple solution to mitigating gradient interference, which substantially improves optimization performance. We provide simple didactic examples and subsequently show significant improvement in optimization for a variety of multi-task supervised learning and reinforcement learning problems. We show that, when some optimization challenges of multi-task learning are alleviated by PCGrad, we can obtain hypothesized benefits in efficiency and asymptotic performance of multi-task settings.

While we studied multi-task supervised learning and multi-task reinforcement learning in this work, we suspect the problem of conflicting gradients to be prevalent in a range of other settings and applications, such as meta-learning, continual learning, multi-goal imitation learning [10], and multi-task problems in natural language processing applications [38]. Due to its simplicity and model-agnostic nature, we expect that applying PCGrad in these domains to be a promising avenue for future investigation. Further, the general idea of gradient surgery may be an important ingredient for alleviating a broader class of optimization challenges in deep learning, such as the challenges in the stability challenges in two-player games [48] and multi-agent optimizations [41]. We believe this work to be a step towards simple yet general techniques for addressing some of these challenges.

## Broader Impact

**Applications and Benefits.** Despite recent success, current deep learning and deep RL methods mostly focus on tackling a single specific task from scratch. Prior methods have proposed methods that can perform multiple tasks, but they often yield comparable or even higher data complexity compared to learning each task individually. Our method enables deep learning systems that mitigate inferences between differing tasks and thus achieves data-efficient multi-task learning. Since our method is general and simple to apply to various problems, there are many possible real-world applications, including but not limited to computer vision systems, autonomous driving, and robotics. For computer vision systems, our method can be used to develop algorithms that enable efficient classification, instance and semantics segmentation and object detection at the same time, which could improve performances of computer vision systems by reusing features obtained from each task and lead to a leap in real-world domains such as autonomous driving. For robotics, there are many situations where multi-task learning is needed. For example, surgical robots are required to perform a wide range of tasks such as stitching and removing tumour from the patient's body. Kitchen robots should be able to complete multiple chores such as cooking and washing dishes at the same time. Hence, our work represents a step towards making multi-task reinforcement learning more applicable to those settings.

**Risks.** However, there are potential risks that apply to all machine learning and reinforcement learning systems including ours, including but not limited to safety, reward specification in RL which is often difficult to acquire in the real world, bias in supervised learning systems due to the composition of training data, and compute/data-intensive training procedures. For example, safety issues arise when autonomous driving cars fail to generalize to out-of-distribution data, which leads to crashing or even hurting people. Moreover, reward specification in RL is generally inaccessible in the real world, making RL unable to scale to real robots. In supervised learning domains, learned models could inherit the bias that exists in the training dataset. Furthermore, training procedures of ML models are generally compute/data-intensive, which cause inequitable access to these models. Our method is not immune to these risks. Hence, we encourage future research to design more robust and safe multi-task RL algorithms that can prevent unsafe behaviors. It is also important to push research in self-supervised and unsupervised multi-task RL in order to resolve the issue of reward specification. For supervised learning, we recommend researchers to publish their trained multi-task learning models to make access to those models equitable to everyone in field and develop new datasets that can mitigate biases and also be readily used in multi-task learning.

## Acknowledgments and Disclosure of Funding

The authors would like to thank Annie Xie for reviewing an earlier draft of the paper, Eric Mitchell for technical guidance, and Aravind Rajeswaran and Deirdre Quillen for helpful discussions. Tianhe Yu is partially supported by Intel Corporation. Saurabh Kumar is supported by an NSF Graduate Research Fellowship and the Stanford Knight Hennessy Fellowship. Abhishek Gupta is supported by an NSF Graduate Research Fellowship. Chelsea Finn is a CIFAR Fellow in the Learning in Machines and Brains program.

## Footnotes

[1]Code is released at `https://github.com/tianheyu927/PCGrad`

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
