[Supplementary Material]

# Appendix

## A   Proofs

### A.1   Proof of Theorem 1

**Theorem 1.** *Assume $\mathcal{L}_1$ and $\mathcal{L}_2$ are convex and differentiable. Suppose the gradient of $\mathcal{L}$ is L-Lipschitz with $L > 0$. Then, the PCGrad update rule with step size $t \leq \frac{1}{L}$ will converge to either (1) a location in the optimization landscape where $\cos(\phi_{12}) = -1$ or (2) the optimal value $\mathcal{L}(\theta^*)$.*

*Proof.* We will use the shorthand $|| \cdot ||$ to denote the $L_2$-norm and $\nabla\mathcal{L} = \nabla_\theta\mathcal{L}$, where $\theta$ is the parameter vector. Following Definition 1 and 4, let $\mathbf{g_1} = \nabla\mathcal{L}_1$, $\mathbf{g_2} = \nabla\mathcal{L}_2$, $\mathbf{g} = \nabla\mathcal{L} = \mathbf{g_1} + \mathbf{g_2}$, and $\phi_{12}$ be the angle between $\mathbf{g_1}$ and $\mathbf{g_2}$.

At each PCGrad update, we have two cases: $cos(\phi_{12}) \geq 0$ or $\cos(\phi_{12}) < 0$.

If $\cos(\phi_{12}) \geq 0$, then we apply the standard gradient descent update using $t \leq \frac{1}{L}$, which leads to a strict decrease in the objective function value $\mathcal{L}(\theta)$ (since it is also convex) unless $\nabla\mathcal{L}(\theta) = 0$, which occurs only when $\theta = \theta^*$ [4].

In the case that $\cos(\phi_{12}) < 0$, we proceed as follows:

Our assumption that $\nabla\mathcal{L}$ is Lipschitz continuous with constant $L$ implies that $\nabla^2\mathcal{L}(\theta) - LI$ is a negative semi-definite matrix. Using this fact, we can perform a quadratic expansion of $\mathcal{L}$ around $\mathcal{L}(\theta)$ and obtain the following inequality:

$$\mathcal{L}(\theta^+) \leq \mathcal{L}(\theta) + \nabla\mathcal{L}(\theta)^T(\theta^+ - \theta) + \frac{1}{2}\nabla^2\mathcal{L}(\theta)||\theta^+ - \theta||^2$$

$$\leq \mathcal{L}(\theta) + \nabla\mathcal{L}(\theta)^T(\theta^+ - \theta) + \frac{1}{2}L||\theta^+ - \theta||^2$$

Now, we can plug in the PCGrad update by letting $\theta^+ = \theta - t \cdot (\mathbf{g} - \frac{\mathbf{g_1} \cdot \mathbf{g_2}}{||\mathbf{g_1}||^2}\mathbf{g_1} - \frac{\mathbf{g_1} \cdot \mathbf{g_2}}{||\mathbf{g_2}||^2}\mathbf{g_2})$. We then get:

$$\mathcal{L}(\theta^+) \leq \mathcal{L}(\theta) + t \cdot \mathbf{g}^T(-\mathbf{g} + \frac{\mathbf{g_1} \cdot \mathbf{g_2}}{||\mathbf{g_1}||^2}\mathbf{g_1} + \frac{\mathbf{g_1} \cdot \mathbf{g_2}}{||\mathbf{g_2}||^2}\mathbf{g_2}) + \frac{1}{2}Lt^2||\mathbf{g} - \frac{\mathbf{g_1} \cdot \mathbf{g_2}}{||\mathbf{g_1}||^2}\mathbf{g_1} - \frac{\mathbf{g_1} \cdot \mathbf{g_2}}{||\mathbf{g_2}||^2}\mathbf{g_2}||^2$$

(Expanding, using the identity $\mathbf{g} = \mathbf{g_1} + \mathbf{g_2}$)

$$= \mathcal{L}(\theta) + t\left(-||\mathbf{g_1}||^2 - ||\mathbf{g_2}||^2 + \frac{(\mathbf{g_1} \cdot \mathbf{g_2})^2}{||\mathbf{g_1}||^2} + \frac{(\mathbf{g_1} \cdot \mathbf{g_2})^2}{||\mathbf{g_2}||^2}\right) + \frac{1}{2}Lt^2||\mathbf{g_1} + \mathbf{g_2}$$
$$- \frac{\mathbf{g_1} \cdot \mathbf{g_2}}{||\mathbf{g_1}||^2}\mathbf{g_1} - \frac{\mathbf{g_1} \cdot \mathbf{g_2}}{||\mathbf{g_2}||^2}\mathbf{g_2}||^2$$

(Expanding further and re-arranging terms)

$$= \mathcal{L}(\theta) - (t - \frac{1}{2}Lt^2)(||\mathbf{g_1}||^2 + ||\mathbf{g_2}||^2 - \frac{(\mathbf{g_1} \cdot \mathbf{g_2})^2}{||\mathbf{g_1}||^2} - \frac{(\mathbf{g_1} \cdot \mathbf{g_2})^2}{||\mathbf{g_2}||^2})$$
$$- Lt^2(\mathbf{g_1} \cdot \mathbf{g_2} - \frac{(\mathbf{g_1} \cdot \mathbf{g_2})^2}{||\mathbf{g_1}||^2||\mathbf{g_2}||^2}\mathbf{g_1} \cdot \mathbf{g_2})$$

(Using the identity $\cos(\phi_{12}) = \frac{\mathbf{g_1} \cdot \mathbf{g_2}}{||\mathbf{g_1}||||\mathbf{g_2}||}$)

$$= \mathcal{L}(\theta) - (t - \frac{1}{2}Lt^2)[(1 - \cos^2(\phi_{12}))||\mathbf{g_1}||^2 + (1 - \cos^2(\phi_{12}))||\mathbf{g_2}||^2]$$
$$- Lt^2(1 - \cos^2(\phi_{12}))||\mathbf{g_1}||||\mathbf{g_2}||\cos(\phi_{12}) \tag{1}$$

(Note that $\cos(\phi_{12}) < 0$ so the final term is non-negative)

Using $t \leq \frac{1}{L}$, we know that $-(1 - \frac{1}{2}Lt) = \frac{1}{2}Lt - 1 \leq \frac{1}{2}L(1/L) - 1 = \frac{-1}{2}$ and $Lt^2 \leq t$.

Plugging this into the last expression above, we can conclude the following:

$$\mathcal{L}(\theta^+) \leq \mathcal{L}(\theta) - \frac{1}{2}t[(1 - \cos^2(\phi_{12}))||\mathbf{g_1}||^2 + (1 - \cos^2(\phi_{12}))||\mathbf{g_2}||^2]$$
$$- t(1 - \cos^2(\phi_{12}))||\mathbf{g_1}||||\mathbf{g_2}|| \cos(\phi_{12})$$
$$= \mathcal{L}(\theta) - \frac{1}{2}t(1 - \cos^2(\phi_{12}))[||\mathbf{g_1}||^2 + 2||\mathbf{g_1}||||\mathbf{g_2}|| \cos(\phi_{12}) + ||\mathbf{g_2}||^2]$$
$$= \mathcal{L}(\theta) - \frac{1}{2}t(1 - \cos^2(\phi_{12}))[||\mathbf{g_1}||^2 + 2\mathbf{g_1} \cdot \mathbf{g_2} + ||\mathbf{g_2}||^2]$$
$$= \mathcal{L}(\theta) - \frac{1}{2}t(1 - \cos^2(\phi_{12}))||\mathbf{g_1} + \mathbf{g_2}||^2$$
$$= \mathcal{L}(\theta) - \frac{1}{2}t(1 - \cos^2(\phi_{12}))||\mathbf{g}||^2$$

If $\cos(\phi_{12}) > -1$, then $\frac{1}{2}t(1 - \cos^2(\phi_{12}))||\mathbf{g}||^2$ will always be positive unless $\mathbf{g} = 0$. This inequality implies that the objective function value strictly decreases with each iteration where $\cos(\phi_{12}) > -1$.

Hence repeatedly applying PCGrad process can either reach the optimal value $\mathcal{L}(\theta) = \mathcal{L}(\theta^*)$ or $\cos(\phi_{12}) = -1$, in which case $\frac{1}{2}t(1 - \cos^2(\phi_{12}))||\mathbf{g}||^2 = 0$. Note that this result only holds when we choose $t$ to be small enough, i.e. $t \leq \frac{1}{L}$.

$\square$

**Corollary 1.** *Assume the $n$ objectives $\mathcal{L}_1, \mathcal{L}_2, ..., \mathcal{L}_n$ are convex and differentiable. Suppose the gradient of $\mathcal{L}$ is Lipschitz continuous with constant $L > 0$. Assume that $\cos(\mathbf{g}, \mathbf{g}^{PC}) \geq \frac{1}{2}$. Then, the PCGrad update rule with step size $t \leq \frac{1}{L}$ will converge to either (1) a location in the optimization landscape where $\cos(\mathbf{g}_i, \mathbf{g}_j) = -1 \forall i, j$ or (2) the optimal value $\mathcal{L}(\theta^*)$.*

*Proof.* Our assumption that $\nabla \mathcal{L}$ is Lipschitz continuous with constant $L$ implies that $\nabla^2 \mathcal{L}(\theta) - LI$ is a negative semi-definite matrix. Using this fact, we can perform a quadratic expansion of $\mathcal{L}$ around $\mathcal{L}(\theta)$ and obtain the following inequality:

$$\mathcal{L}(\theta^+) \leq \mathcal{L}(\theta) + \nabla \mathcal{L}(\theta)^T (\theta^+ - \theta) + \frac{1}{2} \nabla^2 \mathcal{L}(\theta) ||\theta^+ - \theta||^2$$
$$\leq \mathcal{L}(\theta) + \nabla \mathcal{L}(\theta)^T (\theta^+ - \theta) + \frac{1}{2}L ||\theta^+ - \theta||^2$$

Now, we can plug in the PCGrad update by letting $\theta^+ = \theta - t \cdot \mathbf{g}^{PC}$. We then get:

$$\mathcal{L}(\theta^+) \leq \mathcal{L}(\theta) - t \cdot \mathbf{g}^T \mathbf{g}^{PC} + \frac{1}{2}Lt^2 ||\mathbf{g}^{PC}||^2$$

$$\left( \text{Using the assumption that } \cos(\mathbf{g}, \mathbf{g}^{PC}) \geq \frac{1}{2}. \right)$$

$$\leq \mathcal{L}(\theta) - \frac{1}{2}t||\mathbf{g}|| \cdot ||\mathbf{g}^{PC}|| + \frac{1}{2}Lt^2 ||\mathbf{g}^{PC}||^2$$
$$\leq \mathcal{L}(\theta) - \frac{1}{2}t||\mathbf{g}|| \cdot ||\mathbf{g}^{PC}|| + \frac{1}{2}Lt^2 ||\mathbf{g}^{PC}|| \cdot ||\mathbf{g}||$$

Note that $-\frac{1}{2}t||\mathbf{g}|| \cdot ||\mathbf{g}^{PC}|| + \frac{1}{2}Lt^2 ||\mathbf{g}^{PC}|| \cdot ||\mathbf{g}|| \leq 0$ when $t \leq \frac{1}{L}$. Further, when $t < \frac{1}{L}$, $-\frac{1}{2}t||\mathbf{g}|| \cdot ||\mathbf{g}^{PC}|| + \frac{1}{2}Lt^2 ||\mathbf{g}^{PC}|| \cdot ||\mathbf{g}|| = 0$ if and only if $||\mathbf{g}|| = 0$ or $||\mathbf{g}^{PC}|| = 0$.

Hence repeatedly applying PCGrad process can either reach the optimal value $\mathcal{L}(\theta) = \mathcal{L}(\theta^*)$ or a location in the optimization landscape where $\cos(\mathbf{g}_i, \mathbf{g}_j) = -1$ for all pairs of tasks $i, j$. Note that this result only holds when we choose $t$ to be small enough, i.e. $t \leq \frac{1}{L}$. $\square$

**Proposition 1.** *Assume $\mathcal{L}_1$ and $\mathcal{L}_2$ are differentiable but possibly non-convex. Suppose the gradient of $\mathcal{L}$ is Lipschitz continuous with constant $L > 0$. Then, the PCGrad update rule with step size $t \leq \frac{1}{L}$ will converge to either (1) a location in the optimization landscape where $\cos(\phi_{12}) = -1$ or (2) find a $\theta_k$ that is almost a stationary point.*

*Proof.* Following Definition 1 and 4, let $\mathbf{g_{1}}_k = \nabla\mathcal{L}_1$ at iteration $k$, $\mathbf{g_{2}}_k = \nabla\mathcal{L}_2$ at iteration $k$, and $\mathbf{g}_k = \nabla\mathcal{L} = \mathbf{g_{1}}_k + \mathbf{g_{2}}_k$ at iteration $k$, and $\phi_{12,k}$ be the angle between $\mathbf{g_{1}}_k$ and $\mathbf{g_{2}}_k$.

From the proof of Theorem 1, when $\cos(\phi_{12}, k) < 0$ we have:

$$||\mathbf{g}_k||^2 \le \frac{2}{t}\frac{\mathcal{L}(\theta_{k-1}) - \mathcal{L}(\theta_k)}{(1 - \cos^2(\phi_{12,k}))}.$$

Thus, we have:

$$\min_{0 \le k \le K} ||\mathbf{g}_k||^2 \le \frac{1}{K}\sum_{i=0}^{K-1}||\mathbf{g}_i||^2$$

$$\le \frac{2}{Kt}\sum_{i=0}^{K-1}\frac{\mathcal{L}(\theta_{i-1}) - \mathcal{L}(\theta_i)}{(1 - \cos^2(\phi_{12,i}))}$$

If at any iteration, $\cos(\phi_{12,k}) = -1$, then the optimization will stop at that point. If $\forall k \in [0, K]$, $\cos(\phi_{12,k}) \ge \alpha > -1$, then, we have:

$$\min_{0 \le k \le K}||\mathbf{g}_k||^2 \le \frac{2}{K(1 - \alpha^2)t}\sum_{i=0}^{K-1}(\mathcal{L}(\theta_{i-1}) - \mathcal{L}(\theta_i))$$

$$= \frac{2}{K(1 - \alpha^2)t}(\mathcal{L}(\theta_0) - \mathcal{L}(\theta_K))$$

$$\le \frac{2}{K(1 - \alpha^2)t}(\mathcal{L}(\theta_0) - \mathcal{L}^*).$$

where $\mathcal{L}^*$ is the minimal function value.  □

Note that the convergence rate of PCGrad in the non-convex setting largely depends on the value of $\alpha$ and generally how small $\cos(\phi_{12,k})$ is on average.

### A.2 Proof of Theorem 2

**Theorem 2.** *Suppose $\mathcal{L}$ is differentiable and the gradient of $\mathcal{L}$ is Lipschitz continuous with constant $L > 0$. Let $\theta^{MT}$ and $\theta^{PCGrad}$ be the parameters after applying one update to $\theta$ with $\mathbf{g}$ and PCGrad-modified gradient $\mathbf{g}^{PC}$ respectively, with step size $t > 0$. Moreover, assume $\mathbf{H}(\mathcal{L}; \theta, \theta^{MT}) \ge \ell\|\mathbf{g}\|_2^2$ for some constant $\ell \le L$, i.e. the multi-task curvature is lower-bounded. Then $\mathcal{L}(\theta^{PCGrad}) \le \mathcal{L}(\theta^{MT})$ if (a) $\cos\phi_{12} \le -\Phi(\mathbf{g_1}, \mathbf{g_2})$, (b) $\ell \ge \xi(\mathbf{g_1}, \mathbf{g_2})L$, and (c) $t \ge \frac{2}{\ell - \xi(\mathbf{g_1},\mathbf{g_2})L}$.*

*Proof.* Note that $\theta^{MT} = \theta - t \cdot \mathbf{g}$ and $\theta^{PCGrad} = \theta - t(\mathbf{g} - \frac{\mathbf{g_1} \cdot \mathbf{g_2}}{||\mathbf{g_1}||^2}\mathbf{g_1} - \frac{\mathbf{g_1} \cdot \mathbf{g_2}}{||\mathbf{g_2}||^2}\mathbf{g_2})$. Based on the condition that $\mathbf{H}(\mathcal{L}; \theta, \theta^{MT}) \ge \ell\|\mathbf{g}\|_2^2$, we first apply Taylor's Theorem to $\mathcal{L}(\theta^{MT})$ and obtain the following result:

$$\mathcal{L}(\theta^{MT}) = \mathcal{L}(\theta) + \mathbf{g}^T(-t\mathbf{g}) + \int_0^1 (-t\mathbf{g})^T \frac{\nabla^2\mathcal{L}(\theta + a \cdot (-t\mathbf{g}))}{2}(-t\mathbf{g})da$$

$$\ge \mathcal{L}(\theta) + \mathbf{g}^T(-t\mathbf{g}) + t^2 \cdot \frac{1}{2}\ell \cdot \|\mathbf{g}\|_2^2$$

$$= \mathcal{L}(\theta) - t\|\mathbf{g}\|_2^2 + \frac{1}{2}\ell t^2\|\mathbf{g}\|_2^2$$

$$= \mathcal{L}(\theta) + (\frac{1}{2}\ell t^2 - t)\|\mathbf{g}\|_2^2 \qquad (2)$$

where the first inequality follows from Definition 3 and the assumption $\mathbf{H}(\mathcal{L}; \theta, \theta^{MT}) \ge \ell\|\mathbf{g}\|_2^2$. From equation 1, we have the simplified upper bound for $\mathcal{L}(\theta^{PCGrad})$:

$$\mathcal{L}(\theta^{PCGrad}) \le \mathcal{L}(\theta) - (1 - \cos^2\phi_{12})[(t - \frac{1}{2}Lt^2)\cdot(\|\mathbf{g_1}\|_2^2 + \|\mathbf{g_1}\|_2^2) + Lt^2\|\mathbf{g_1}\|_2\|\mathbf{g_2}\|_2\cos\phi_{12}]$$

$$(3)$$

Apply Equation 2 and Equation 3 and we have the following inequality:

$$\mathcal{L}(\theta^{\mathrm{MT}}) - \mathcal{L}(\theta^{\mathrm{PCGrad}}) \geq \mathcal{L}(\theta) + (\frac{1}{2}\ell t^2 - t)\|\mathbf{g}\|_2^2 - \mathcal{L}(\theta)$$

$$+ (1 - \cos^2\phi_{12})[(t - \frac{1}{2}Lt^2)(\|\mathbf{g_1}\|_2^2 + \|\mathbf{g_2}\|_2^2) + Lt^2\|\mathbf{g_1}\|_2\|\mathbf{g_2}\|_2\cos\phi_{12}]$$

$$= (\frac{1}{2}\ell t^2 - t)\|\mathbf{g_1} + \mathbf{g_2}\|_2^2 + (1 - \cos^2\phi_{12})\Big[(t - \frac{1}{2}Lt^2)\cdot(\|\mathbf{g_1}\|_2^2 + \|\mathbf{g_2}\|_2^2) + Lt^2\|\mathbf{g_1}\|_2\|\mathbf{g_2}\|_2\cos\phi_{12}\Big]$$

$$= \left(\frac{1}{2}\|\mathbf{g_1} + \mathbf{g_2}\|_2^2\ell - \frac{1 - \cos^2\phi_{12}}{2}(\|\mathbf{g_1}\|_2^2 + \|\mathbf{g_2}\|_2^2 - 2\|\mathbf{g_1}\|_2\|\mathbf{g_2}\|_2\cos\phi_{12})L\right)t^2$$

$$- \left((\|\mathbf{g_1}\|_2^2 + \|\mathbf{g_2}\|_2^2)\cos^2\phi_{12} + 2\|\mathbf{g_1}\|_2\|\mathbf{g_2}\|_2\cos\phi_{12}\right)t$$

$$= \left(\frac{1}{2}\|\mathbf{g_1} + \mathbf{g_2}\|_2^2\ell - \frac{1 - \cos^2\phi_{12}}{2}\|\mathbf{g_1} - \mathbf{g_2}\|_2^2 L\right)t^2$$

$$- \left((\|\mathbf{g_1}\|_2^2 + \|\mathbf{g_2}\|_2^2)\cos^2\phi_{12} + 2\|\mathbf{g_1}\|_2\|\mathbf{g_2}\|_2\cos\phi_{12}\right)t$$

$$= t\cdot\left[\left(\frac{1}{2}\|\mathbf{g_1} + \mathbf{g_2}\|_2^2\ell - \frac{1 - \cos^2\phi_{12}}{2}(\|\mathbf{g_1} - \mathbf{g_2}\|_2^2)L\right)t\right.$$

$$\left. - \left((\|\mathbf{g_1}\|_2^2 + \|\mathbf{g_2}\|_2^2)\cos^2\phi_{12} + 2\|\mathbf{g_1}\|_2\|\mathbf{g_2}\|_2\cos\phi_{12}\right)\right] \tag{4}$$

Since $\cos\phi_{12} \leq -\Phi(\mathbf{g_1}, \mathbf{g_2}) = -\frac{2\|\mathbf{g_1}\|_2\|\mathbf{g_2}\|_2}{\|\mathbf{g_1}\|_2^2 + \|\mathbf{g_2}\|_2^2}$ and $\ell \geq \xi(\mathbf{g_1}, \mathbf{g_2}) = \frac{(1-\cos^2\phi_{12})(\|\mathbf{g_1} - \mathbf{g_2}\|_2^2)}{\|\mathbf{g_1} + \mathbf{g_2}\|_2^2}L$, we have

$$\frac{1}{2}\|\mathbf{g_1} + \mathbf{g_2}\|_2^2\ell - \frac{1 - \cos^2\phi_{12}}{2}\|\mathbf{g_1} - \mathbf{g_2}\|_2^2 L \geq 0$$

and

$$(\|\mathbf{g_1}\|_2^2 + \|\mathbf{g_2}\|_2^2)\cos^2\phi_{12} + 2\|\mathbf{g_1}\|_2\|\mathbf{g_2}\|_2\cos\phi_{12} \geq 0.$$

By the condition that $t \geq \frac{2}{\ell - \xi(\mathbf{g_1}, \mathbf{g_2})L} = \frac{2}{\ell - \frac{(1-\cos^2\phi_{12})\|\mathbf{g_1} - \mathbf{g_2}\|_2^2}{\|\mathbf{g_1} + \mathbf{g_2}\|_2^2}L}$ and monotonicity of linear functions, we have the following:

$$\mathcal{L}(\theta^{\mathrm{MT}}) - \mathcal{L}(\theta^{\mathrm{PCGrad}}) \geq \left[\left(\frac{1}{2}\|\mathbf{g_1} + \mathbf{g_2}\|_2^2\ell - \frac{1 - \cos^2\phi_{12}}{2}\cdot\|\mathbf{g_1} - \mathbf{g_2}\|_2^2 L\right)\cdot\frac{2}{\ell - \frac{(1-\cos^2\phi_{12})\|\mathbf{g_1} - \mathbf{g_2}\|_2^2}{\|\mathbf{g_1} + \mathbf{g_2}\|_2^2}L}\right.$$

$$- \left((\|\mathbf{g_1}\|_2^2 + \|\mathbf{g_2}\|_2^2)\cos^2\phi_{12} + 2\|\mathbf{g_1}\|_2\|\mathbf{g_2}\|_2\cos\phi_{12})\right]\cdot t$$

$$= [\|\mathbf{g_1} + \mathbf{g_2}\|_2^2\cdot\left(\ell - \frac{(1 - \cos^2\phi_{12})\cdot\|\mathbf{g_1} - \mathbf{g_2}\|_2^2}{\|\mathbf{g_1} + \mathbf{g_2}\|_2^2}L\right)\cdot\frac{1}{\ell - \frac{(1-\cos^2\phi_{12})\|\mathbf{g_1} - \mathbf{g_2}\|_2^2}{\|\mathbf{g_1} + \mathbf{g_2}\|_2^2}L}$$

$$- \left((\|\mathbf{g_1}\|_2^2 + \|\mathbf{g_2}\|_2^2)\cos^2\phi_{12} + 2\|\mathbf{g_1}\|_2\|\mathbf{g_2}\|_2\cos\phi_{12})\right]\cdot t$$

$$= \left[\|\mathbf{g_1} + \mathbf{g_2}\|_2^2 - ((\|\mathbf{g_1}\|_2^2 + \|\mathbf{g_2}\|_2^2)\cos^2\phi_{12} + 2\|\mathbf{g_1}\|_2\|\mathbf{g_2}\|_2\cos\phi_{12})\right]\cdot t$$

$$= \left[\|\mathbf{g_1}\|_2^2 + \|\mathbf{g_2}\|_2^2 + 2\|\mathbf{g_1}\|_2\|\mathbf{g_2}\|_2\cos\phi_{12} - ((\|\mathbf{g_1}\|_2^2 + \|\mathbf{g_2}\|_2^2)\cos^2\phi_{12} + 2\|\mathbf{g_1}\|_2\|\mathbf{g_2}\|_2\cos\phi_{12})\right]\cdot t$$

$$= (1 - \cos^2\phi_{12})(\|\mathbf{g_1}\|_2^2 + \|\mathbf{g_2}\|_2^2)\cdot t$$

$$\geq 0$$

$\square$

## A.3 PCGrad: Sufficient and Necessary Conditions for Loss Improvement

Beyond the sufficient conditions shown in Theorem 2, we also present the sufficient and necessary conditions under which PCGrad achieves lower loss after one gradient update in Theorem 3 in the two-task setting.

**Theorem 3.** *Suppose $\mathcal{L}$ is differentiable and the gradient of $\mathcal{L}$ is Lipschitz continuous with constant $L > 0$. Let $\theta^{MT}$ and $\theta^{PCGrad}$ be the parameters after applying one update to $\theta$ with $\mathbf{g}$ and PCGrad-modified gradient $\mathbf{g}^{PC}$ respectively, with step size $t > 0$. Moreover, assume $\mathbf{H}(\mathcal{L}; \theta, \theta^{MT}) \geq \ell\|\mathbf{g}\|_2^2$ for some constant $\ell \leq L$, i.e. the multi-task curvature is lower-bounded. Then $\mathcal{L}(\theta^{PCGrad}) \leq \mathcal{L}(\theta^{MT})$ if and only if*

- $-\Phi(\mathbf{g_1}, \mathbf{g_2}) \leq \cos\phi_{12} < 0$

- $\ell \leq \xi(\mathbf{g_1}, \mathbf{g_2})L$

- $0 < t \leq \frac{(\|\mathbf{g_1}\|_2^2 + \|\mathbf{g_2}\|_2^2)\cos^2\phi_{12} + 2\|\mathbf{g_1}\|_2\|\mathbf{g_2}\|_2\cos\phi_{12}}{\frac{1}{2}\|\mathbf{g_1}+\mathbf{g_2}\|_2^2\ell - \frac{1-\cos^2\phi_{12}}{2}(\|\mathbf{g_1}-\mathbf{g_2}\|_2^2)L}$

*or*

- $\cos\phi_{12} \leq -\Phi(\mathbf{g_1}, \mathbf{g_2})$

- $\ell \geq \xi(\mathbf{g_1}, \mathbf{g_2})L$

- $t \geq \frac{(\|\mathbf{g_1}\|_2^2 + \|\mathbf{g_2}\|_2^2)\cos^2\phi_{12} + 2\|\mathbf{g_1}\|_2\|\mathbf{g_2}\|_2\cos\phi_{12}}{\frac{1}{2}\|\mathbf{g_1}+\mathbf{g_2}\|_2^2\ell - \frac{1-\cos^2\phi_{12}}{2}(\|\mathbf{g_1}-\mathbf{g_2}\|_2^2)L}.$

*Proof.* To show the necessary conditions, from Equation 4, all we need is

$$
\begin{aligned}
t \cdot [&(\frac{1}{2}\|\mathbf{g_1} + \mathbf{g_2}\|_2^2\ell - \frac{1-\cos^2\phi_{12}}{2}(\|\mathbf{g_1} - \mathbf{g_2}\|_2^2)L)t \\
&- ((\|\mathbf{g_1}\|_2^2 + \|\mathbf{g_2}\|_2^2)\cos^2\phi_{12} + 2\|\mathbf{g_1}\|_2\|\mathbf{g_2}\|_2\cos\phi_{12})] \geq 0
\end{aligned}
\tag{5}
$$

Since $t \geq 0$, it reduces to show

$$
\begin{aligned}
(\frac{1}{2}\|\mathbf{g_1} + \mathbf{g_2}\|_2^2\ell &- \frac{1-\cos^2\phi_{12}}{2}(\|\mathbf{g_1} - \mathbf{g_2}\|_2^2)L)t \\
&- ((\|\mathbf{g_1}\|_2^2 + \|\mathbf{g_2}\|_2^2)\cos^2\phi_{12} + 2\|\mathbf{g_1}\|_2\|\mathbf{g_2}\|_2\cos\phi_{12}) \geq 0
\end{aligned}
\tag{6}
$$

For Equation 6 to hold while ensuring that $t \geq 0$, there are two cases:

- $\frac{1}{2}\|\mathbf{g_1} + \mathbf{g_2}\|_2^2\ell - (1 - \cos^2\phi_{12})(\|\mathbf{g_1}\|_2^2 + \|\mathbf{g_2}\|_2^2)L \geq 0$,
  $(\|\mathbf{g_1}\|_2^2 + \|\mathbf{g_2}\|_2^2)\cos^2\phi_{12} + 2\|\mathbf{g_1}\|_2\|\mathbf{g_2}\|_2\cos\phi_{12} \geq 0$,
  $t \geq \frac{(\|\mathbf{g_1}\|_2^2 + \|\mathbf{g_2}\|_2^2)\cos^2\phi_{12} + 2\|\mathbf{g_1}\|_2\|\mathbf{g_2}\|_2\cos\phi_{12}}{\frac{1}{2}\|\mathbf{g_1}+\mathbf{g_2}\|_2^2\ell - \frac{1-\cos^2\phi_{12}}{2}(\|\mathbf{g_1}-\mathbf{g_2}\|_2^2)L}$

- $\frac{1}{2}\|\mathbf{g_1} + \mathbf{g_2}\|_2^2\ell - (1 - \cos^2\phi_{12})(\|\mathbf{g_1}\|_2^2 + \|\mathbf{g_2}\|_2^2)L \leq 0$,
  $(\|\mathbf{g_1}\|_2^2 + \|\mathbf{g_2}\|_2^2)\cos^2\phi_{12} + 2\|\mathbf{g_1}\|_2\|\mathbf{g_2}\|_2\cos\phi_{12} \leq 0$,
  $t \geq \frac{(\|\mathbf{g_1}\|_2^2 + \|\mathbf{g_2}\|_2^2)\cos^2\phi_{12} + 2\|\mathbf{g_1}\|_2\|\mathbf{g_2}\|_2\cos\phi_{12}}{\frac{1}{2}\|\mathbf{g_1}+\mathbf{g_2}\|_2^2\ell - \frac{1-\cos^2\phi_{12}}{2}(\|\mathbf{g_1}-\mathbf{g_2}\|_2^2)L}$

, which can be simplified to

- $\cos\phi_{12} \leq -\frac{2\|\mathbf{g_1}\|_2\|\mathbf{g_2}\|_2}{\|\mathbf{g_1}\|_2^2 + \|\mathbf{g_2}\|_2^2} = -\Phi(\mathbf{g_1}, \mathbf{g_2})$,
  $\ell \geq \frac{(1-\cos^2\phi_{12})(\|\mathbf{g_1}\|_2^2 + \|\mathbf{g_2}\|_2^2)}{\|\mathbf{g_1}+\mathbf{g_2}\|_2^2}L = \xi(\mathbf{g_1}, \mathbf{g_2})$,
  $t \geq \frac{(\|\mathbf{g_1}\|_2^2 + \|\mathbf{g_2}\|_2^2)\cos^2\phi_{12} + 2\|\mathbf{g_1}\|_2\|\mathbf{g_2}\|_2\cos\phi_{12}}{\frac{1}{2}\|\mathbf{g_1}+\mathbf{g_2}\|_2^2\ell - \frac{1-\cos^2\phi_{12}}{2}(\|\mathbf{g_1}-\mathbf{g_2}\|_2^2)L}$

- $-\frac{2\|\mathbf{g_1}\|_2\|\mathbf{g_2}\|_2}{\|\mathbf{g_1}\|_2^2 + \|\mathbf{g_2}\|_2^2} = -\Phi(\mathbf{g_1}, \mathbf{g_2}) \leq \cos\phi_{12} < 0$,
  $\ell \leq \frac{(1-\cos^2\phi_{12})(\|\mathbf{g_1}\|_2^2 + \|\mathbf{g_2}\|_2^2)}{\|\mathbf{g_1}+\mathbf{g_2}\|_2^2}L = \xi(\mathbf{g_1}, \mathbf{g_2})$,
  $0 < t \leq \frac{(\|\mathbf{g_1}\|_2^2 + \|\mathbf{g_2}\|_2^2)\cos^2\phi_{12} + 2\|\mathbf{g_1}\|_2\|\mathbf{g_2}\|_2\cos\phi_{12}}{\frac{1}{2}\|\mathbf{g_1}+\mathbf{g_2}\|_2^2\ell - \frac{1-\cos^2\phi_{12}}{2}(\|\mathbf{g_1}-\mathbf{g_2}\|_2^2)L}.$

The sufficient conditions hold as we can plug the conditions to RHS of Equation 6 and achieve non-negative result. $\square$

## A.4 Convergence of PCGrad with Momentum-Based Gradient Descent

In this subsection, we show convergence of PCGrad with momentum-based methods, which is more aligned with our practical implementation. In our analysis, we consider the heavy ball method [43] as follows:

$$\theta_{k+1} \leftarrow \theta_k - \alpha_k \nabla \mathcal{L}(\theta_k) + \beta_k(\theta_k - \theta_{k-1})$$

where $k$ denotes the $k$-th step and $\alpha_k$ and $\beta_k$ are step sizes for the gradient and momentum at step $k$ respectively. We now present our theorem.

**Theorem 4.** *Assume $\mathcal{L}_1$ and $\mathcal{L}_2$ are $\mu_1$- and $\mu_2$-strongly convex and also $L_1$- and $L_2$-smooth respectively where $\mu_1, \mu_2, L_1, L_2 > 0$. Define $\phi_{12}^k$ as the angle between two task gradients $\mathbf{g}_1(\theta_k)$ and $\mathbf{g}_2(\theta_k)$ and define $R_k = \frac{\|\mathbf{g}_1(\theta_k)\|}{\|\mathbf{g}_2(\theta_k)\|}$. Denote $\mu_k = (1 - \cos \phi_{12}^k/R_k)\mu_1 + (1 - \cos \phi_{12}^k \cdot R_k)\mu_2$ and $L_k = (1 - \cos \phi_{12}^k/R_k)L_1 + (1 - \cos \phi_{12}^k \cdot R_k)L_2$ Then, the PCGrad update rule of the heavy ball method with step sizes $\alpha_k = \frac{4}{\sqrt{L_k} + \sqrt{\mu_k}}$ and $\beta_k = \max\{|1 - \sqrt{\alpha_k \mu_k}|, |1 - \sqrt{\alpha_k L_k}|\}^2$ will converge linearly to either (1) a location in the optimization landscape where $\cos(\phi_{12}^k) = -1$ or (2) the optimal value $\mathcal{L}(\theta^*)$.*

*Proof.* We first observe that the PCGrad-modified gradient $\mathbf{g}^{\mathrm{PC}}$ has the following identity:

$$
\begin{aligned}
\mathbf{g}^{\mathrm{PC}} &= \mathbf{g} - \frac{\mathbf{g_1} \cdot \mathbf{g_2}}{\|\mathbf{g_1}\|^2}\mathbf{g_1} - \frac{\mathbf{g_1} \cdot \mathbf{g_2}}{\|\mathbf{g_2}\|^2}\mathbf{g_2} \\
&= (1 - \frac{\mathbf{g_1} \cdot \mathbf{g_2}}{\|\mathbf{g_1}\|^2})\mathbf{g_1} + (1 - \frac{\mathbf{g_1} \cdot \mathbf{g_2}}{\|\mathbf{g_2}\|^2})\mathbf{g_2} \\
&= (1 - \frac{\mathbf{g_1} \cdot \mathbf{g_2}}{\|\mathbf{g_1}\|\|\mathbf{g_2}\|}\frac{\|\mathbf{g_2}\|}{\|\mathbf{g_1}\|})\mathbf{g_1} + (1 - \frac{\mathbf{g_1} \cdot \mathbf{g_2}}{\|\mathbf{g_1}\|\|\mathbf{g_2}\|}\frac{\|\mathbf{g_1}\|}{\|\mathbf{g_2}\|})\mathbf{g_2} \\
&= (1 - \cos \phi_{12}/R)\mathbf{g_1} + (1 - \cos \phi_{12} \cdot R)\mathbf{g_2}.
\end{aligned}
\tag{7}
$$

Applying Equation 7, we can write the PCGrad update rule of the heavy ball method in matrix form as follows:

$$
\begin{aligned}
\left\|\begin{bmatrix} \theta_{k+1} - \theta^* \\ \theta_k - \theta^* \end{bmatrix}\right\|_2 &= \left\|\begin{bmatrix} \theta_k + \beta_k(\theta_k - \theta_{k-1}) - \theta^* \\ \theta_k - \theta^* \end{bmatrix} - \alpha_k \begin{bmatrix} \mathbf{g}^{\mathrm{PC}}(\theta_k) \\ 0 \end{bmatrix}\right\|_2 \\
&= \left\|\begin{bmatrix} \theta_k + \beta_k(\theta_k - \theta_{k-1}) - \theta^* \\ \theta_k - \theta^* \end{bmatrix} \right. \\
&\quad \left. -\alpha_k \begin{bmatrix} (1 - \cos \phi_{12}^k/R_k)\mathbf{g_1}(\theta_k) + (1 - \cos \phi_{12}^k \cdot R_k)\mathbf{g_2}(\theta_k) \\ 0 \end{bmatrix}\right\|_2 \\
&= \left\|\begin{bmatrix} \theta_k + \beta_k(\theta_k - \theta_{k-1}) - \theta^* \\ \theta_k - \theta^* \end{bmatrix} \right. \\
&\quad \left. -\alpha_k \begin{bmatrix} \left[(1 - \cos \phi_{12}^k/R_k)\nabla^2\mathcal{L}_1(z_k) + (1 - \cos \phi_{12}^k \cdot R_k)\nabla^2\mathcal{L}_2(z_k')\right](\theta_k - \theta^*) \\ 0 \end{bmatrix}\right\|_2
\end{aligned}
$$

for some $z_k, z_k'$ on the line segment between $\theta_k$ and $\theta^*$

$$
\begin{aligned}
&= \left\|\begin{bmatrix} (1 + \beta_k)I - \alpha_k H_k & -\beta_k I \\ I & 0 \end{bmatrix}\begin{bmatrix} \theta_k - \theta^* \\ \theta_{k-1} - \theta^* \end{bmatrix}\right\|_2 \\
&\leq \left\|\begin{bmatrix} (1 + \beta_k)I - \alpha_k H_k & -\beta_k I \\ I & 0 \end{bmatrix}\right\|_2 \left\|\begin{bmatrix} \theta_k - \theta^* \\ \theta_{k-1} - \theta^* \end{bmatrix}\right\|_2
\end{aligned}
$$

where $H_k = (1 - \cos \phi_{12}^k/R_k)\nabla^2\mathcal{L}_1(z_k) + (1 - \cos \phi_{12}^k \cdot R_k)\nabla^2\mathcal{L}_2(z_k')$.

By strong convexity and smoothness of $\mathcal{L}_1$ and $\mathcal{L}_2$, we have the eigenvalues of $\nabla^2\mathcal{L}_1(z_k)$ are between $\mu_1$ and $L_1$. Similarly, the eigenvalues of $\nabla^2\mathcal{L}_2(z_k')$ are between $\mu_2$ and $L_2$. Thus the eigenvalues of $H_k$ are between $\mu_k = (1 - \cos \phi_{12}^k/R_k)\mu_1 + (1 - \cos \phi_{12}^k \cdot R_k)\mu_2$ and $L_k = (1 - \cos \phi_{12}^k/R_k)L_1 + (1 - \cos \phi_{12}^k \cdot R_k)L_2$ [20]. Hence following Lemma 3.1 in [54], we have

$$
\left\|\begin{bmatrix} (1 + \beta_k)I - \alpha_k H_k & -\beta_k I \\ I & 0 \end{bmatrix}\right\|_2 \leq \max\{|1 - \sqrt{\alpha_k \mu_k}|, |1 - \sqrt{\alpha_k L_k}|\}.
$$

Thus we have

$$\left\| \begin{bmatrix} \theta_{k+1} - \theta^* \\ \theta_k - \theta^* \end{bmatrix} \right\|_2 \le \max\{|1 - \sqrt{\alpha_k \mu_k}|, |1 - \sqrt{\alpha_k L_k}|\} \left\| \begin{bmatrix} \theta_k - \theta^* \\ \theta_{k-1} - \theta^* \end{bmatrix} \right\|_2$$

$$= \frac{\sqrt{\kappa_k} - 1}{\sqrt{\kappa_k} + 1} \left\| \begin{bmatrix} \theta_k - \theta^* \\ \theta_{k-1} - \theta^* \end{bmatrix} \right\|_2 \qquad (8)$$

$$\le \left\| \begin{bmatrix} \theta_k - \theta^* \\ \theta_{k-1} - \theta^* \end{bmatrix} \right\|_2$$

where $\kappa_k = \frac{L_k}{\mu_k}$ and Equation 8 follows from substitution $\alpha_k = \frac{4}{\sqrt{L_k} + \sqrt{\mu_k}}$. Hence PCGrad with heavy ball method converges linearly if $\cos \phi_{12}^k \ne -1$. $\qquad\square$

# B  Empirical Objective-Wise Evaluations of PCGrad

In this section, we visualize the per-task training loss and validation loss curves respectively on NYUv2. The goal of measuring objective-wise performance is to study the convergence of PCGrad in practice, particularly amidst the possibility of slow convergence due to cosine similarities near -1, as discussed in Section 2.4.

We show the objective-wise evaluation results on NYUv2 in Figure 5. For evaluations on NYUv2, PCGrad + MTAN attains similar training convergence rate compared to MTAN in three tasks in NYUv2 while converging faster and achieving lower validation loss in 2 out of 3 tasks. Note that in task 0 of the NYUv2 dataset, both methods seem to overfit, suggesting a better regularization scheme for this domain.

In general, these results suggest that PCGrad has a regularization effect on supervised multi-task learning, rather than an improvement on optimization speed or convergence. We hypothesize that this regularization is caused by PCGrad leading to greater sharing of representations across tasks, such that the supervision for one task better regularizes the training of another. This regularization effect seems notably different from the effect of PCGrad on reinforcement learning problems, where PCGrad dramatically improves training performance. This suggests that multi-task supervised learning and multi-task reinforcement learning problems may have distinct challenges.

Figure 5: Empirical objective-wise evaluations on NYUv2. On the top row, we show the objective-wise training learning curves and on the bottom row, we show the objective-wise validation learning curves. PCGrad+MTAN converges with a similar rate compared to MTAN in training and for validation losses, PCGrad+MTAN converges faster and obtains a lower final validation loss in two out of three tasks. This result corroborate that in practice, PCGrad does not exhibit the potential slow convergence problem shown in Theorem 1.

# C  Practical Details of PCGrad on Multi-Task and Goal-Conditioned Reinforcement Learning

In our experiments, we apply PCGrad to the soft actor-critic (SAC) algorithm [24], an off-policy RL method. In SAC, we employ a Q-learning style gradient to compute the gradient of the Q-function

network, $Q_\phi(s, a, z_i)$, often known as the critic, and a reparameterization-style gradient to compute the gradient of the policy network $\pi_\theta(a|s, z_i)$, often known as the actor. For sampling, we instantiate a set of replay buffers $\{\mathcal{D}_i\}_{\mathcal{T}_i \sim p(\mathcal{T})}$. Training and data collection are alternated throughout training. During a data collection step, we run the policy $\pi_\theta$ on all the tasks $\mathcal{T}_i \sim p(\mathcal{T})$ to collect an equal number of paths for each task and store the paths of each task $\mathcal{T}_i$ into the corresponding replay buffer $\mathcal{D}_i$. At each training step, we sample an equal amount of data from each replay buffer $\mathcal{D}_i$ to form a stratified batch. For each task $\mathcal{T}_i \sim p(\mathcal{T})$, the parameters of the critic $\theta$ are optimized to minimize the soft Bellman residual:

$$J_Q^{(i)}(\phi) = \mathbb{E}_{(s_t, a_t, z_i) \sim \mathcal{D}_i} \left[ Q_\phi(s_t, a_t, z_i) - (r(s_t, a_t, z_i) + \gamma V_{\bar{\phi}}(s_{t+1}, z_i)) \right], \tag{9}$$

$$V_{\bar{\phi}}(s_{t+1}, z_i) = \mathbb{E}_{a_{t+1} \sim \pi_\theta} \left[ Q_{\bar{\phi}}(s_{t+1}, a_{t+1}, z_i) - \alpha \log \pi_\theta(a_{t+1}|s_{t+1}, z_i) \right], \tag{10}$$

where $\gamma$ is the discount factor, $\bar{\phi}$ are the delayed parameters, and $\alpha$ is a learnable temperature that automatically adjusts the weight of the entropy term. For each task $\mathcal{T}_i \sim p(\mathcal{T})$, the parameters of the policy $\pi_\theta$ are trained to minimize the following objective

$$J_\pi^{(i)}(\theta) = \mathbb{E}_{s_t \sim \mathcal{D}_i} \left[ \mathbb{E}_{a_t \sim \pi_\theta(a_t|s_t, z_i))} \left[ \alpha \log \pi_\theta(a_t|s_t, z_i) - Q_\phi(s_t, a_t, z_i) \right] \right]. \tag{11}$$

We compute $\nabla_\phi J_Q^{(i)}(\phi)$ and $\nabla_\theta J_\pi^{(i)}(\theta)$ for all $\mathcal{T}_i \sim p(\mathcal{T})$ and apply PCGrad to both following Algorithm 1.

In the context of SAC specifically, we also propose to learn the temperature $\alpha$ for adjusting entropy of the policy on a per-task basis. This allows the method to control the entropy of the multi-task policy per-task. The motivation is that if we use a single learnable temperature for adjusting entropy of the multi-task policy $\pi_\theta(a|s, z_i)$, SAC may stop exploring once all easier tasks are solved, leading to poor performance on tasks that are harder or require more exploration. To address this issue, we propose to learn the temperature on a per-task basis as mentioned in Section 3, i.e. using a parametrized model to represent $\alpha_\psi(z_i)$. This allows the method to control the entropy of $\pi_\theta(a|s, z_i)$ per-task. We optimize the parameters of $\alpha_\psi(z_i)$ using the same constrained optimization framework as in [24].

When applying PCGrad to goal-conditioned RL, we represent $p(\mathcal{T})$ as a distribution of goals and let $z_i$ be the encoding of a goal. Similar to the multi-task supervised learning setting discussed in Section 3, PCGrad may be combined with various architectures designed for multi-task and goal-conditioned RL [19, 14], where PCGrad operates on the gradients of shared parameters, leaving task-specific parameters untouched.

## D    2D Optimization Landscape Details

To produce the 2D optimization visualizations in Figure 1, we used a parameter vector $\theta = [\theta_1, \theta_2] \in \mathbb{R}^2$ and the following task loss functions:

$$\mathcal{L}_1(\theta) = 20 \log(\max(|.5\theta_1 + \tanh(\theta_2)|, 0.000005))$$
$$\mathcal{L}_2(\theta) = 25 \log(\max(|.5\theta_1 - \tanh(\theta_2) + 2|, 0.000005))$$

The multi-task objective is $\mathcal{L}(\theta) = \mathcal{L}_1(\theta) + \mathcal{L}_2(\theta)$. We initialized $\theta = [0.5, -3]$ and performed 500,000 gradient updates to minimize $\mathcal{L}$ using the Adam optimizer with learning rate 0.001. We compared using Adam for each update to using Adam in conjunction with the PCGrad method presented in Section 2.3.

## E    Additional Multi-Task Supervised Learning Results

We present our multi-task supervised learning results on MultiMNIST and CityScapes here.

**MultiMNIST.**    Following the same set-up of Sener and Koltun [53], for each image, we sample a different one uniformly at random. Then we put one of the image on the top left and the other one on the bottom right. The two tasks in the multi-task learning problem are to classify the digits on the top left (task-L) and bottom right (task-R) respectively. We construct such 60K examples. We combine PCGrad with the same backbone architecture used in [53] and compare its performance to Sener and Koltun [53] by running the open-sourced code provided in [53]. As shown in Table 4, PCGrad results 0.13% and 0.55% improvement over [53] in left and right digit accuracy respectively.

|  | left digit | right digit |
|---|---|---|
| Sener and Koltun [53] | 96.45 | 94.95 |
| PCGrad (ours) | **96.58** | **95.50** |

Table 4: MultiMNIST results. PCGrad achieves improvements over the approach by Sener and Koltun [53] in both left and right digit classfication accuracy.

**CityScapes.** The CityScapes dataset [12] contains 19 classes of street-view images resized to $128 \times 256$. There are two tasks in this dataset: semantic segmentation and depth estimation. Following the setup in Liu et al. [33], we pair the depth estimation task with semantic segmentation using the coarser 7 categories instead of the finer 19 classes in the original CityScapes dataset. Similar to NYUv2 evaluations described in Section 5, we also combine PCGrad with MTAN [33] and compare it to a range of methods discussed in Appendix J.1. For the combination of PCGrad and MTAN, we only use equal weighting as discussed in [33] as we find it working well in practice. We present the results in Table 5. As shown in Table 5, PCGrad + MTAN outperforms MTAN in three out of four scores while obtaining the top scores in both mIoU abd pixel accuracy for the semantic segmentation task, suggesting the effectiveness of PCGrad on realistic image datasets. We also provide the full results including three different weighting schemes in Table 7 in Appendix J.4.

| #P. | Architecture | Segmentation (Higher Better) | | Depth (Lower Better) | |
|---|---|---|---|---|---|
| | | mIoU | Pix Acc | Abs Err | Rel Err |
| 2 | One Task | 51.09 | 90.69 | 0.0158 | 34.17 |
| 3.04 | STAN | 51.90 | 90.87 | 0.0145 | **27.46** |
| 1.75 | Split, Wide | 50.17 | 90.63 | 0.0167 | 44.73 |
| 2 | Split, Deep | 49.85 | 88.69 | 0.0180 | 43.86 |
| 3.63 | Dense | 51.91 | 90.89 | **0.0138** | 27.21 |
| $\approx 2$ | Cross-Stitch [39] | 50.08 | 90.33 | 0.0154 | 34.49 |
| 1.65 | MTAN | 53.04 | 91.11 | 0.0144 | 33.63 |
| 1.65 | PCGrad+MTAN (Ours) | **53.59** | **91.45** | 0.0171 | 31.34 |

Table 5: We present the 7-class semantic segmentation and depth estimation results on CityScapes dataset. We use #P to denote the number of parameters of the network. We use box and bold text to highlight the method that achieves the best validation score for each task. As seen in the results, PCGrad+MTAN with equal weights outperforms MTAN with equal weights in three out of four scores while achieving the top score both scores in the segmentation task.

# F    Goal-Conditioned Reinforcement Learning Results

For our goal-conditioned RL evaluation, we adopt the goal-conditioned robotic pushing task with a Sawyer robot where the goals are represented as the concatenations of the initial positions of the puck to be pushed and the its goal location, both of which are uniformly sampled (details in Appendix J.3). We also apply the temperature adjustment strategy as discussed in Section 3 to predict the temperature for entropy term given the goal. We summarize the results in the plot second from right in Figure 3. PCGrad with SAC achieves better performance in terms of average distance to the goal position, while the vanilla SAC agent is struggling to successfully accomplish the task. This suggests that PCGrad is able to ease the RL optimization problem also when the task distribution is continuous.

# G    Comparison to CosReg

We compare PCGrad to a prior method CosReg [55], which adds a regularization term to force the cosine similarity between gradients of two different tasks to stay $0$. PCGrad achieves much better average success rate in MT10 benchmark as shown in Figure 7. Hence, while it's important to reduce interference between tasks, it's also crucial to keep the task gradients that enjoy positive cosine similarities in order to ensure sharing across tasks.

Figure 6: We present the goal-conditioned RL results. PCGrad outperforms vanilla SAC in terms of both average distance the goal and data efficiency.

Figure 7: Comparison between PCGrad and CosReg [55]. PCGrad outperforms CosReg, suggesting that we should both reduce the interference and keep shared structure across tasks.

## H  Ablation study on the task order

As stated on line 4 in Algorithm 1, we sample the tasks from the batch and randomly shuffle the order of the tasks before performing the update steps in PCGrad. With random shuffling, we make PCGrad symmetric w.r.t. the task order in expectation. In Figure 8, we observe that PCGrad with a random task order achieves better performance between PCGrad with a fixed task order in the setting of MT50 where the number of tasks is large and the conflicting gradient phenomenon is much more likely to happen.

## I  Combining PCGrad with other architectures

In this subsection, we test whether PCGrad can improve performances when combined with more methods. In Table 6, we find that PCGrad does improve the performance in all four metrics of the three tasks on the NYUv2 dataset when combined with Cross-Stitch [39] and Dense. In Figure 9, we also show that PCGrad + Multi-head SAC outperforms Multi-head SAC on its own. These results suggest that PCGrad can be flexibly combined with any multi-task learning architectures to further improve performance.

Figure 8: Ablation study on using a fixed task order during PCGrad. PCGrad with a random task order does significantly better PCGrad with a fixed task order in MT50 benchmark.

| Method | Segmentation | Depth | Surface Normal | |
|---|---|---|---|---|
| | mIoU | Abs Err | Angle Distance | Within $11.25°$ |
| Cross-Stitch | 15.69 | 0.6277 | 32.69 | 21.63 |
| Cross-Stitch + PCGrad | **18.14** | **0.5805** | **31.38** | **21.75** |
| Dense | 16.48 | 0.6282 | 31.68 | 21.73 |
| Dense + PCGrad | **18.08** | **0.5850** | **30.17** | **23.29** |

Table 6: We show the performances of PCGrad combined with other methods on three-task learning on the NYUv2 dataset, where PCGrad further improves the results of prior multi-task learning architectures.

## J   Experiment Details

### J.1   Multi-Task Supervised Learning Experiment Details

For all the multi-task supervised learning experiments, PCGrad converges within 12 hours on a NVIDIA TITAN RTX GPU while the vanilla models without PCGrad converge within 8 hours. PCGrad consumes at most 10 GB memory on GPU while the vanilla method consumes 6GB on GPU among all experiments.

For our CIFAR-100 multi-task experiment, we adopt the architecture used in [47], which is a convolutional neural network that consists of 3 convolutional layers with $160$ $3 \times 3$ filters each layer and 2 fully connected layers with 320 hidden units. As for experiments on the NYUv2 dataset, we follow [33] to use SegNet [1] as the backbone architecture.

We use five algorithms as baselines in the CIFAR-100 multi-task experiment: **task specific-1-fc** [46]: a convolutional neural network shared across tasks except that each task has a separate last fully-connected layer, **task specific-1-fc** [46] : all the convolutional layers shared across tasks with separate fully-connected layers for each task, **cross stitch-all-fc** [40]: one convolutional neural network per task along with cross-stitch units to share features across tasks, **routing-all-fc + WPL** [47]: a network that employs a trainable router trained with multi-agent RL algorithm (WPL) to select trainable functions for each task, **independent**: training separate neural networks for each task.

For comparisons on the NYUv2 dataset, we consider 5 baselines: **Single Task, One Task**: the vanilla SegNet used for single-task training, **Single Task, STAN** [33]: the single-task version of MTAN as mentioned below, **Multi-Task, Split, Wide / Deep** [33]: the standard SegNet shared for all three tasks except that each task has a separate last layer for final task-specific prediction with two variants **Wide** and **Deep** specified in [33], **Multi-Task Dense**: a shared network followed by separate task-specific networks, **Multi-Task Cross-Stitch** [40]: similar to the baseline used in CIFAR-100 experiment but with SegNet as the backbone, **MTAN** [33]: a shared network with a soft-attention module for each task.

### J.2   Multi-Task Reinforcement Learning Experiment Details

Our reinforcement learning experiments all use the SAC [24] algorithm as the base algorithm, where the actor and the critic are represented as 6-layer fully-connected feedforward neural networks for all

Figure 9: We show the comparison between Multi-head SAC and Multi-head SAC + PCGrad on MT10. Multi-head SAC + PCGrad outperforms Multi-head SAC, suggesting that PCGrad can improves the performance of multi-headed architectures in the multi-task RL settings.

Figure 10: The 50 tasks of MT50 from Meta-World [61]. MT10 is a subset of these tasks, which includes reach, push, pick & place, open drawer, close drawer, open door, press button top, open window, close window, and insert peg inside.

methods. The numbers of hidden units of each layer of the neural networks are 160, 300 and 200 for MT10, MT50 and goal-conditioned RL respectively. For the multi-task RL experiments, PCGrad + SAC converges in 1 day (5M simulation steps) and 5 days (20M simulation steps) on the MT10 and MT50 benchmarks respectively on a NVIDIA TITAN RTX GPU while vanilla SAC converges in 12 hours and 3 days on the two benchmarks respectively. PCGrad + SAC consumes 1 GB and 6 GB memory on GPU on the MT10 and MT50 benchmarks respectively while the vanilla SAC consumes 0.5 GB and 3 GB respectively.

In the case of multi-task reinforcement learning, we evaluate our algorithm on the recently proposed Meta-World benchmark [61]. This benchmark includes a variety of simulated robotic manipulation tasks contained in a shared, table-top environment with a simulated Sawyer arm (visualized in Fig. 10). In particular, we use the multi-task benchmarks MT10 and MT50, which consists of the 10 tasks and 50 tasks respectively depicted in Fig. 10 that require diverse strategies to solve them, which makes them difficult to optimize jointly with a single policy. Note that MT10 is a subset of MT50. At each

data collection step, we collect 600 samples for each task, and at each training step, we sample 128 datapoints per task from corresponding replay buffers. We measure success according to the metrics used in the Meta-World benchmark where the reported the success rates are averaged across tasks. For all methods, we apply the temperature adjustment strategy as discussed in Section 3 to learn a separate alpha term per task as the task encoding in MT10 and MT50 is just a one-hot encoding.

On the multi-task and goal-conditioned RL domain, we apply PCGrad to the vanilla SAC algorithm with task encoding as part of the input to the actor and the critic as described in Section 3 and compare PCGrad to the vanilla **SAC** without PCGrad and training actors and critics for each task individually (**Independent**).

## J.3    Goal-conditioned Experiment Details

We use the pushing environment from the Meta-World benchmark [61] as shown in Figure 10. In this environment, the table spans from $[-0.4, 0.2]$ to $[0.4, 1.0]$ in the 2D space. To construct the goals, we sample the intial positions of the puck from the range $[-0.2, 0.6]$ to $[0.2, 0.7]$ on the table and the goal positions from the range $[-0.2, 0.85]$ to $[0.2, 0.95]$ on the table. The goal is represented as a concatenation of the initial puck position and the goal position. Since in the goal-conditioned setting, the task distribution is continuous, we sample a minibatch of 9 goals and 128 samples per goal at each training iteration and also sample 600 samples per goal in the minibatch at each data collection step.

## J.4    Full CityScapes and NYUv2 Results

We provide the full comparison on the CityScapes and NYUv2 datasets in Table 7 and Table 8 respectively.

| #P. | Architecture | Weighting | Segmentation (Higher Better) | | Depth (Lower Better) | |
|---|---|---|---|---|---|---|
| | | | mIoU | Pix Acc | Abs Err | Rel Err |
| 2 | One Task | n.a. | 51.09 | 90.69 | 0.0158 | 34.17 |
| 3.04 | STAN | n.a. | 51.90 | 90.87 | 0.0145 | 27.46 |
| 1.75 | Split, Wide | Equal Weights | 50.17 | 90.63 | 0.0167 | 44.73 |
| | | Uncert. Weights [28] | **51.21** | **90.72** | **0.0158** | 44.01 |
| | | DWA, $T = 2$ | 50.39 | 90.45 | 0.0164 | **43.93** |
| 2 | Split, Deep | Equal Weights | **49.85** | 88.69 | 0.0180 | 43.86 |
| | | Uncert. Weights [28] | 48.12 | 88.68 | **0.0169** | **39.73** |
| | | DWA, $T = 2$ | 49.67 | **88.81** | 0.0182 | 46.63 |
| 3.63 | Dense | Equal Weights | **51.91** | 90.89 | 0.0138 | 27.21 |
| | | Uncert. Weights [28] | 51.89 | **91.22** | 0.0134 | 25.36 |
| | | DWA, $T = 2$ | 51.78 | 90.88 | 0.0137 | 26.67 |
| ≈2 | Cross-Stitch [39] | Equal Weights | 50.08 | 90.33 | 0.0154 | 34.49 |
| | | Uncert. Weights [28] | 50.31 | 90.43 | **0.0152** | **31.36** |
| | | DWA, $T = 2$ | **50.33** | **90.55** | 0.0153 | 33.37 |
| 1.65 | MTAN | Equal Weights | 53.04 | 91.11 | **0.0144** | 33.63 |
| | | Uncert. Weights [28] | 53.86 | 91.10 | 0.0144 | 35.72 |
| | | DWA, $T = 2$ | 53.29 | 91.09 | 0.0144 | 34.14 |
| 1.65 | PCGrad+MTAN (Ours) | Equal Weights | 53.59 | 91.45 | 0.0171 | **31.34** |

Table 7: We present the 7-class semantic segmentation and depth estimation results on CityScapes dataset. We use #P to denote the number of parameters of the network, and the best performing variant of each architecture is highlighted in bold. We use box to highlight the method that achieves the best validation score for each task. As seen in the results, PCGrad+MTAN with equal weights outperforms MTAN with equal weights in three out of four scores while achieving the top score in pixel accuracy across all methods.

| Type | #P. | Architecture | Weighting | Segmentation (Higher Better) | | Depth (Lower Better) | | Surface Normal Angle Distance (Lower Better) | | Within $t°$ (Higher Better) | | |
|---|---|---|---|---|---|---|---|---|---|---|---|---|
| | | | | mIoU | Pix Acc | Abs Err | Rel Err | Mean | Median | 11.25 | 22.5 | 30 |
| Single Task | 3 | One Task | n.a. | 15.10 | 51.54 | 0.7508 | 0.3266 | 31.76 | 25.51 | 22.12 | 45.33 | 57.13 |
| | 4.56 | STAN† | n.a. | 15.73 | 52.89 | 0.6935 | 0.2891 | 32.09 | 26.32 | 21.49 | 44.38 | 56.51 |
| Multi Task | 1.75 | Split, Wide | Equal Weights | 15.89 | 51.19 | 0.6494 | 0.2804 | 33.69 | 28.91 | 18.54 | 39.91 | 52.02 |
| | | | Uncert. Weights* | 15.86 | 51.12 | **0.6040** | 0.2570 | **32.33** | **26.62** | **21.68** | **43.59** | **55.36** |
| | | | DWA†, $T = 2$ | **16.92** | **53.72** | 0.6125 | **0.2546** | 32.34 | 27.10 | 20.69 | 42.73 | 54.74 |
| | 2 | Split, Deep | Equal Weights | 13.03 | 41.47 | 0.7836 | 0.3326 | 38.28 | 36.55 | 9.50 | 27.11 | 39.63 |
| | | | Uncert. Weights* | **14.53** | 43.69 | 0.7705 | 0.3340 | **35.14** | **32.13** | **14.69** | **34.52** | **46.94** |
| | | | DWA†, $T = 2$ | 13.63 | **44.41** | **0.7581** | **0.3227** | 36.41 | 34.12 | 12.82 | 31.12 | 43.48 |
| | 4.95 | Dense | Equal Weights | 16.06 | 52.73 | 0.6488 | 0.2871 | 33.58 | 28.01 | 20.07 | 41.50 | 53.35 |
| | | | Uncert. Weights* | **16.48** | **54.40** | 0.6282 | 0.2761 | **31.68** | **25.68** | **21.73** | **44.58** | **56.65** |
| | | | DWA†, $T = 2$ | 16.15 | 54.35 | **0.6059** | **0.2593** | 32.44 | 27.40 | 20.53 | 42.76 | 54.27 |
| | ≈3 | Cross-Stitch‡ | Equal Weights | 14.71 | 50.23 | 0.6481 | 0.2871 | 33.56 | 28.58 | 20.08 | 40.54 | 51.97 |
| | | | Uncert. Weights* | 15.69 | 52.60 | 0.6277 | 0.2702 | 32.69 | 27.26 | 21.63 | 42.84 | 54.45 |
| | | | DWA†, $T = 2$ | **16.11** | **53.19** | **0.5922** | **0.2611** | **32.34** | **26.91** | **21.81** | **43.14** | **54.92** |
| | 1.77 | MTAN† | Equal Weights | 17.72 | 55.32 | 0.5906 | 0.2577 | 31.44 | 25.37 | 23.17 | 45.65 | 57.48 |
| | | | Uncert. Weights* | 17.67 | **55.61** | 0.5927 | 0.2592 | **31.25** | 25.57 | 22.99 | **45.83** | **57.67** |
| | | | DWA†, $T = 2$ | 17.15 | 54.97 | 0.5956 | 0.2569 | 31.60 | 25.46 | 22.48 | 44.86 | 57.24 |
| | 1.77 | MTAN† + PCGrad (ours) | Uncert. Weights* | 20.17 | 56.65 | 0.5904 | 0.2467 | 30.01 | 24.83 | 22.28 | 46.12 | 58.77 |

Table 8: We present the full results on three tasks on the NYUv2 dataset: 13-class semantic segmentation, depth estimation, and surface normal prediction results. #P shows the total number of network parameters. We highlight the best performing combination of multi-task architecture and weighting in bold. The top validation scores for each task are annotated with boxes. The symbols indicate prior methods: *: [28], †: [33], ‡: [40]. Performance of other methods taken from [33].