[Reviews · NeurIPS 2020]

Review 1

Summary and Contributions: This manuscript proposes a new technique for certain classes multi-task learning (MTL) problems - PCGrad. The authors formalize the three conditions in MTL that they plan to address, and attempt to show empirically and theoretically that their approach addresses those conditions.

Strengths: Identifies and formalizes at least two challenging conditions for MTL; Proposes a novel approach; Evaluates the approach.

Weaknesses: Limited significant ML contributions. Limited experiments.

Correctness: Yes - the claims seem to be correct, and the methodology is correct.

Clarity: Yes.

Relation to Prior Work: Yes.

Reproducibility: Yes

Additional Feedback:


Review 2

Summary and Contributions: The paper proposes a gradient-based method for tackling multi-task learning problem, in which "conflicting" gradients are detected and altered so that such situations would not harm the common objective. The authors also provide the theoretical and experimental results to support their claim that the proposed method is more data-efficient and better in performance.

Strengths: - The paper is well motivated by the problem of gradient conflict in the optimization perspective and is well written. - The problem is well demonstrated in Appendix B, justifying the assumptions. - The experiments are performed well with several tasks and datasets, showing that PCGrad outperforms baselines in most cases.

Weaknesses: - The results except for Table 1 are not totally better than the baselines, and even worse in the depth task in Table 4. - The results in Table 2 for PCGrad alone is also worse than [44], in combination with other cases when PCGrad is worse, raising the questions about the usefulness of PCGrad "alone" in practice. While it is useful to see that PCGrad is indeed model-agnostic since it deals directly with gradients, one might argue that PCGrad itself, alone, is sometimes useful but not always the case. So maybe the authors need to discuss further the reason why, and also there's a need to conduct more experiments in which PCGrad is combined with more methods, not just only one for each table, to convincingly prove the usefulness of PCGrad with broader practicality.

Correctness: The authors carry out well the motivation, methodology, and experiments. However, as addressed in the weaknesses section, some might argue that the experiments are not entirely convincing and need extra work to make the paper better.

Clarity: The paper is well written and easy to follow.

Relation to Prior Work: The authors discussed in detail about prior work, separated into 4 sections: multi-task learning in general, architectural and divide-and-conquer solutions, optimization-based methods, and methods in the area of continual learning.

Reproducibility: Yes

Additional Feedback: In the main theorem 1, the assumptions of convexity of the losses are probably too strong. While many people might have done the same thing, it is worth discussing in greater detail about how this might differ in practice centering around this theorem. This would make the connection between theory and application of the paper closely related and hence more consistent. Post rebuttal: I would like to thank the authors for their effort in responding to my questions and would like to keep the previous score.


Review 3

Summary and Contributions: The paper address the problem of multi-task learning. It hypothesizes that in contrast (or in addition) to other challenges that make this problem harder than single-task learning, is the "conflicting gradient" problem. They authors propose a method to project each task gradient onto the normal plan of the other tasks alleviate the conflicts and update the gradient.

Strengths: The paper tackles one of the most interesting topics - multi-task learning. It hypothesizes a reasonable theory about a problem in applying single-task learning SGD to multi-task, and follow up with an algorithm. The algorithm and assumptions undergo in-depth technical analysis, convex optimization settings proofs for sanity checks and multiple domain experiments. The paper is also well written.

Weaknesses: There was no special reference to run-time and computation complexity although it seems like it should not be too much of a problem. I would have like to see more evidence of the problem. For example, the number of gradient conflicts or/and the angle between the gradient for several known problems averaged for each epoch training (or something similar)

Correctness: The empirical evaluation is solid.

Clarity: very well written

Relation to Prior Work: yes

Reproducibility: Yes

Additional Feedback:


Review 4

Summary and Contributions: This paper proposed projecting conflicting gradients (PCGrad) to solve the problem of conflicting gradient in multitask learning. Experiments on computer vision tasks and reinforcement learning tasks verifies the effectiveness of the proposed method.

Strengths: The paper is easy to follow, and the performance of the algorithm is really impressive.

Weaknesses: 1. Though the toy example in Fig. 1 is quite intuitive and proof of convergence under convex case is provided, the paper misses an important part: how severe is the conflicting gradient problem is in practical tasks? I suggest the authors to draw some plots of the cosine similarity between the gradients from different losses. I also went through the appendix, but failed to find such plots. This is my main concern. 2. Minor: It would be better to provide comparison with more methods, e.g., [1], [2], etc.. 3. The output of Alg. 1 is not deterministic due to random sampling. Though I understand it's a workaround for practical use, it makes the generalization of Theorem 1 and 2 to the case n>2 (the number of task loss functions) non-trivial. It would be better if the authors can provide more analysis on this. [1] GradNorm: Gradient Normalization for Adaptive Loss Balancing in Deep Multitask Networks, Zhao Chen et al., ICML 2018 [2] Multi-Task Learning as Multi-Objective Optimization, Ozan Sener et al., NeurIPS 2018 -----------------Post Rebuttal----------------- After reading the authors' response, I changed my rating to 6. (1) I did miss Corollary 1in the appendix. Thanks for pointing out. (2) Q1: There is only one figure in the appendix to justify the existence of the problem, which gives me an impression of cherry-picking. (3) Q2: Comparison with these important baselines in the main text and main experiments (Sec. 5.1) was missing. I suggest the authors to make substantial improvement to provide a more solid foundation for the existence of the problem, as well as more convincing experiments in the future version.

Correctness: Yes, both the claims and methods are correct.

Clarity: The paper is well written. It is easy to follow.

Relation to Prior Work: Yes, the paper clearly discussed the difference.

Reproducibility: Yes

Additional Feedback:

[Author Response · NeurIPS 2020]

| Method | Segmentation | Depth | Surface Normal | |
|---|---|---|---|---|
| | mIoU | Abs Err | Angle Distance | Within $11.25^\circ$ |
| Cross-Stitch | 15.69 | 0.6277 | 32.69 | 21.63 |
| Cross-Stitch + PCGrad | **18.14** | **0.5805** | **31.38** | **21.75** |
| Dense | 16.48 | 0.6282 | 31.68 | 21.73 |
| Dense + PCGrad | **18.08** | **0.5850** | **30.17** | **23.29** |

Table 1: As requested by reviewer 2, we show new experiments with PCGrad combined with other methods on three-task learning on the NYUv2 dataset.

Figure 1: As requested by reviewer 2, we show the comparison between Multi-head SAC and Multi-head SAC + PCGrad on MT10.

We thank all the reviewers for the constructive feedback. We will incorporate the valuable suggestions in the revised
version. We have addressed all of the comments below:

**R4, Q1: I suggest the authors to draw some plots of the cosine similarity between the gradients from different**
**losses?** We already have these plots in the paper, see Figure 4 in Appendix B. We will move these plots into the main
paper. The middle plot in Figure 4 visualizes the cosine similarity between the gradients of two Meta-World tasks.
Based on the plot, we can see that the cosine similarity between task gradients are negative more than 50% of the
iterations and condition (a) in Thm 2 holds for most of the time that both tasks haven't been solved. This suggests that
the conflicting gradients problem is indeed a challenge in practice.

**R4, Q2: comparison with more methods, e.g. GradNorm [1] and Sener et al [2]?** We already compare to [1] in
the rightmost plot in Figure 3. We already compare to [2] in Table 3 and Table 5 in Appendix F. PCGrad outperforms
both [1] and [2]. We will make sure these comparisons appear prominently in the main paper.

**R4, Q3: The output of Alg. 1 is not deterministic due to random sampling ... the generalization of Theorem 1**
**and 2 to the case $n > 2$ (the number of task loss functions) is non-trivial?** We use random sampling to make our
gradient projection procedure symmetric to the order of projection in expectation. We already provided analysis of the
convergence of Theorem 1 where we have more than 2 tasks in Corollary 1 in Appendix A.1. We will also include the
generalization of Theorem 2 with $n > 2$ in the final paper.

**R1, Q1: Limited significant ML contributions and experiments?** Our paper identifies three conditions that lead to
poor MTL performance and proposes a new MTL algorithm that tackles them. In the paper, we provide the theory
that motivates the practical algorithm and observe strong empirical results on 8 challenging benchmarks (5 supervised
MTL benchmarks, 2 MTRL benchmarks and 1 goal-conditioned RL benchmark). Hence, we think the paper meets the
standard of NeurIPS papers in terms of ML contributions and experiments.

**R2, Q1: The results except for Table 1 are not totally better than the baselines, and even worse in the depth task**
**in Table 4?** PPCGrad is better than the baselines in four out of five supervised learning and all three RL experiments.
In Table 1, PCGrad outperforms all prior methods in segmentation and depth, and in 4/5 metrics of surface normal
prediction. In Table 4 as R2 said, PCGrad does not outperform the baseline in the depth task. This is potentially because
CityScapes dataset only contains 2 tasks and the two tasks may not have much interference.
We also emphasize that the results are not cherry-picked. We report results on all benchmarks that we tried, and seven
of the eight benchmarks were proposed in prior works.

**R2, Q2: PCGrad alone...Conduct more experiments in which PCGrad is combined with more methods?** PCGrad
is considerably simpler than routing networks [44], as it doesn't require an RL optimization and requires only a few
lines of code on top of a vanilla multi-task learning model.
We added more experiments to test whether PCGrad can improve performances when combined with more methods. In
Table 1 of the response, we find that PCGrad does improve the performance in all four metrics of the three tasks on the
NYUv2 dataset when combined with Cross-Stitch and Dense. In Figure 1 of the response, we also show that PCGrad +
Multi-head SAC outperforms Multi-head SAC on its own. We will include the additional experiments in the final paper.

**R2, Q3: the assumptions of convexity of the losses are probably too strong in Theorem 1?** In Proposition 1 in
Appendix A.1, we show that PCGrad converges to a stationary point when the losses are nonconvex, which is more
connected to practice.

**R3, Q1: There was no special reference to run-time and computation complexity?** For all the multi-task supervised
learning experiments, PCGrad converges within 12 hours on a NVIDIA TITAN RTX GPU while the vanilla models
without PCGrad converge within 8 hours. PCGrad consumes at most 10 GB memory on GPU while the vanilla method
consumes 6GB on GPU among all experiments. For the multi-task RL experiments, PCGrad + SAC converges in 1
day (5M simulation steps) and 5 days (20M simulation steps) on the MT10 and MT50 benchmarks respectively on a
NVIDIA TITAN RTX GPU while vanilla SAC converges in 12 hours and 3 days on the two benchmarks respectively.
PCGrad + SAC consumes 1 GB and 6 GB memory on GPU on the MT10 and MT50 benchmarks respectively while the
vanilla SAC consumes 0.5 GB and 3 GB respectively. We will include the discussion in the final paper.

[Meta-Review · NeurIPS 2020]

The paper proposes a gradient-based method for tackling multi-task learning problem, in which "conflicting" gradients are detected and altered so that such situations would not harm the common objective. The reviewers came to a consensus that the proposed technology is sound and the paper has its value to the NeurIPS community. However, the reviewers also gave some constructive feedback, e.g., moving the comparisons with important baselines to the main body of the paper, instead of placing them in the appendix. We hope the authors could follow these suggestions when preparing their final version of the paper,